# Phosphoproteome Dynamics of *Streptomyces rimosus* during Submerged Growth and Antibiotic Production

Ela Šarić,[a] Gerry A. Quinn,[a,b] Nicolas Nalpas,[c] Tina Paradžik,[a] Saša Kazazić,[a] Želimira Filić,[a] Maja Šemanjski,[c] Paul Herron,[d] Iain Hunter,[d] Boris Maček,[c] Dušica Vujaklija[a]

[a]Division of Physical Chemistry, Institute Ruđer Bošković, Zagreb, Croatia
[b]Centre for Molecular Biosciences, Ulster University, Coleraine, United Kingdom
[c]Proteome Center Tübingen, University of Tübingen, Tübingen, Germany
[d]Strathclyde Institute of Pharmacy and Biomedical Sciences, University of Strathclyde, Glasgow, United Kingdom

**ABSTRACT** *Streptomyces rimosus* is an industrial streptomycete, best known as a producer of oxytetracycline, one of the most widely used antibiotics. Despite the significant contribution of *Streptomyces* species to the pharmaceutical industry, most omics analyses have only been conducted on the model organism *Streptomyces coelicolor*. In recent years, protein phosphorylation on serine, threonine, and tyrosine (Ser, Thr, and Tyr, respectively) has been shown to play a crucial role in the regulation of numerous cellular processes, including metabolic changes leading to antibiotic production and morphological changes. In this study, we performed a comprehensive quantitative (phospho)proteomic analysis during the growth of *S. rimosus* under conditions of oxytetracycline production and pellet fragmentation. Liquid chromatography-tandem mass spectrometry (LC-MS/MS) analysis combined with phosphopeptide enrichment detected a total of 3,725 proteins, corresponding to 45.6% of the proteome and 417 phosphorylation sites from 230 phosphoproteins. Significant changes in abundance during three distinct growth phases were determined for 494 proteins and 98 phosphorylation sites. Functional analysis revealed changes in phosphorylation events of proteins involved in important cellular processes, including regulatory mechanisms, primary and secondary metabolism, cell division, and stress response. About 80% of the phosphoproteins detected during submerged growth of *S. rimosus* have not yet been reported in streptomycetes, and 55 phosphoproteins were not reported in any prokaryote studied so far. This enabled the creation of a unique resource that provides novel insights into the dynamics of (phospho)proteins and reveals many potential regulatory events during antibiotic production in liquid culture of an industrially important bacterium.

**IMPORTANCE** *Streptomyces rimosus* is best known as a primary source of oxytetracycline (OTC). The significant global market value of OTC highlights the need for a better understanding of the regulatory mechanisms that lead to production of this antibiotic. Our study provides, for the first time, a detailed insight into the dynamics of (phospho)proteomic profiles during growth and antibiotic production in liquid culture of *S. rimosus*. Significant changes in protein synthesis and phosphorylation have been revealed for a number of important cellular proteins during the growth stages that coincide with OTC production and morphological changes of this industrially important bacterium. Most of these proteins have not been detected in previous studies. Therefore, our results significantly expand the insight into phosphorylation events associated with important cellular processes and antibiotic production; they also greatly increase the phosphoproteome of streptomycetes and contribute with newly discovered phosphoproteins to the database of prokaryotic phosphoproteomes. This can consequently lead to the design of novel research directions in elucidation of the complex regulatory network in *Streptomyces*.

Address correspondence to Boris Maček, boris.macek@uni-tuebingen.de, or Dušica Vujaklija, vujaklij@irb.hr.
The authors declare no conflict of interest.
[This article was published on 12 September 2022 with errors in the Fig. 5 legend. The legend was updated in the current version, posted on 23 September 2022.]

**KEYWORDS** *Streptomyces rimosus*, oxytetracycline production, pellet fragmentation, proteome, phosphoproteome, peptide dimethylation labeling, oxytetracycline

Streptomycetes are multicellular Gram-positive bacteria found in soil and other niches such as marine and freshwater ecosystems. This genus has a complex morphological differentiation resembling that of fungi, with great genetic potential to produce antibiotics, anticancer drugs, immunomodulators, and other bioactives of significant medical or industrial value (1–3). Growth of streptomycetes on solid media starts with spore germination, hyphal development, and branching to form a vegetative mycelium, followed by differentiation into aerial hyphae that eventually septate into hydrophobic spore chains (4). Manteca et al. (5) provided a detailed insight into pellet development and physiological differentiation, associated with production of specialized metabolites, while growing *Streptomyces coelicolor* A3(2) in liquid cultures (6, 7). Streptomyces have large genomes, typically from 6 to 12 Mb, with >7,000 protein-coding genes (8, 9) reflecting their developmental complexity, potential for adaptation to changing environmental niches, and capacity to produce a wide range of antibiotics and other bioactive metabolites. Consequently, a large proportion of their genes have regulatory roles whose functions coordinate their responses to the changing environment (7). More than 900 genes (~12%) are predicted to have a regulatory role in *S. coelicolor* (10).

Protein phosphorylation plays a key regulatory role in cell signaling, gene expression, and cell differentiation in all domains of life. In bacteria, phosphoproteomic studies have been mostly performed on model and pathogenic species (11). Very few phosphoproteomes of industrially important bacteria have been reported, e.g., glutamate-producing *Corynebacterium glutamicum* (12); *Lactoccocus lactis*, widely employed in the food industry (13, 14); and the erythromycin-producing actinobacterium, *Saccharopolyspora erythrea* (13). In the genus *Streptomyces*, phosphoproteomic analyses have only been performed for the model organism *S. coelicolor*, which does not produce commercial antibiotics. The first global phosphoproteomic study of *S. coelicolor* reported 46 phosphorylation sites on 40 proteins identified in cell extracts of submerged mycelia harvested during mid-exponential growth (15). Subsequent extensive phosphoproteome analysis during *S. coelicolor* differentiation on solid medium identified 127 phosphoproteins with 289 phosphorylation sites, including sporulation factors, transcriptional regulators, protein kinases, and other regulatory proteins involved in morphological differentiation (16). More recently, quantitative (phospho)proteome analysis of *S. coelicolor* revealed phosphorylation dynamics in many proteins involved in differentiation and activation of secondary metabolism (17), leading to the conclusion that manipulating the phosphorylation of target proteins may improve secondary metabolite production or activate silent biosynthetic gene clusters (BCGs) in industrial streptomycetes. In agreement with previous studies showing that expression of *S. coelicolor* phosphorylated regulatory protein (AfsR) increased antibiotic production even when introduced into heterologous systems (18, 19).

Since phosphoproteomic analysis has never been performed on commercially important streptomycetes, the objective of this study was to gain comprehensive insight into the dynamics of phosphorylation events of the industrially important microorganism, *Streptomyces rimosus*, which has become one of the best genetically described streptomycetes (20). *S. rimosus* is best known as the primary source of oxytetracycline (OTC), a significant clinical natural product used against a wide range of Gram-positive and Gram-negative pathogens, mycoplasmas, chlamydiae, rickettsiae, and protozoan parasites (21, 22). The significant global market value of OTC (>$10^8$ kg annually) highlights the need for improved productivity by commercial strains (23). *S. rimosus* ATCC 10970, also known as G7 or R7, is the wild-type and parental strain of the many OTC production strains (20). Its draft genome sequence was completed in 2013 (24) and recently resequenced by Pacific Biosciences (PacBio) (GenBank accession number CP048261). The *S. rimosus* genome is 9.64 Mb (including a 0.29 Mb giant linear plasmid), which is within the range of other published streptomycete genome sequences. Recently, an uncommon set of telomeres of the linear *S. rimosus* G7 chromosome has been revealed (25).

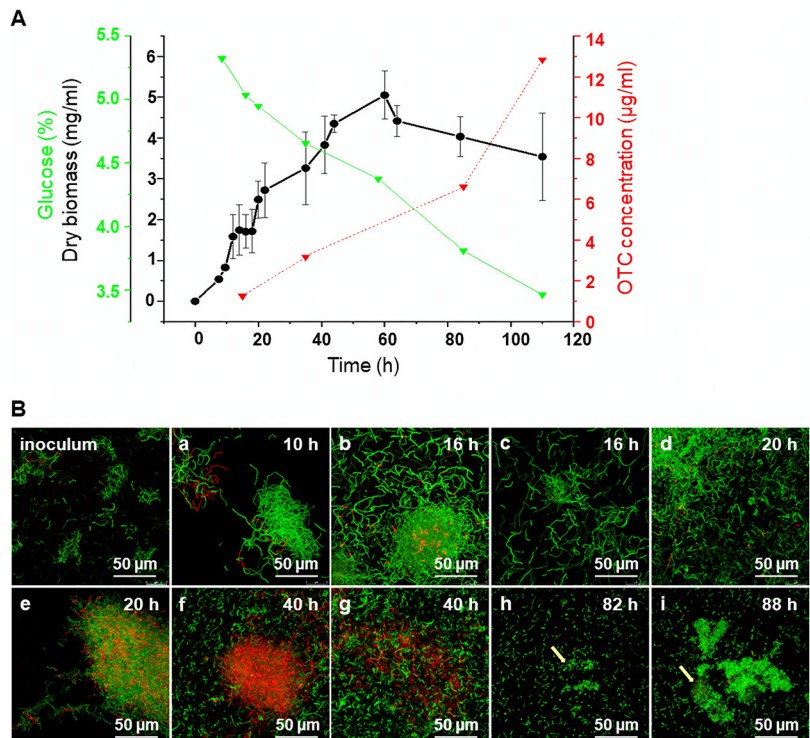

**FIG 1** Growth and mycelial fragmentation of *S. rimosus* G7 in submerged culture, from early exponential to late stationary phase. (A) Growth curve based on dry weight accumulation (sampling time of biomass is indicated by arrows; error bars indicate standard deviations). This figure also shows increase in OTC (red) and decrease in glucose (green). (B) Mycelial fragmentation at various time points was followed by confocal fluorescence microscopy after SYTO 9 and PI staining (green and red hyphae represent live and dead cells, respectively, while arrows point to the fragments embedded in the extracellular matrix).

The genome is estimated to have a total of 8,166 protein-coding genes (263 encoded by the plasmid). Among the protein-coding genes, 32 are predicted to be serine/threonine protein kinases (STPKs). A previous study confirmed protein tyrosine phosphorylation in the *S. rimosus* wild-type strain (26).

Here, we determined the dynamics of (phospho)proteome profiles in *S. rimosus* during three very distinct stages of bacterial growth. Most of the detected phosphoproteins (155) have been unreported and provide new insights into the posttranslational regulation during antibiotic synthesis and morphological changes that culminate in massive pellet fragmentation. Functional analysis of identified phosphoproteins revealed that they participate in regulatory mechanisms, primary and secondary metabolism, cell division, stress response, and others. Our results greatly increased the phosphoproteome of streptomycetes and provided important information for studying the role of this posttranslational modification in metabolic changes leading to antibiotic synthesis.

## RESULTS AND DISCUSSION

**Mycelial morphology changes during the growth arrest phase and coincides with oxytetracycline production.** Large-scale OTC production is performed with submerged *S. rimosus* cultures (5). Since protein phosphorylation has a key role in regulating various cellular processes, including production of specialized metabolites (27, 28), we studied the dynamics of STY protein phosphorylation within the context of antibiotic production during the submerged growth of the parental strain, *S. rimosus* G7, which grows quickly and in a highly dispersed form. Qualitative and quantitative analyses of the (phospho)proteome were performed using liquid medium standardized for low and high producers. These conditions were employed by industry to screen for strains

with higher antibiotic production at the lab scale. As depicted in Fig. 1A, *S. rimosus* G7 exhibited, in submerged culture, a multiphasic growth curve, first observed in *Streptomyces hygroscopicus* (29). During the exponential growth phase, bacteria enter into transient growth arrest associated with activation of antibiotic synthesis and resistance (29, 30), with a decrease in ribosomal proteins (31) and a stress response (32). Neumann et al. also observed growth arrest in *Streptomyces griseus*, suggesting that this phenomenon is probably a general property of streptomycetes (33). Manteca et al. reported that mycelial differentiation leads to antibiotic production in *S. coelicolor* during that phase and showed results of proteome analysis (5, 34), while Nieselt et al. observed a major metabolic shift prior to antibiotic production (35). Growth arrest (Fig. 1A) was also observed in *S. rimosus* (~16 h to 20 h; SI). To the best of our knowledge, it has never been reported for this species, even though this bacterium has been widely studied over several decades (20). Since our aim was to obtain a better insight into phosphorylation events that might be crucial for OTC production, this observation determined the choice of the first growth point for the collection of bacterial biomasses. In addition, biomass was collected at the beginning of the stationary phase (SII) and in late stationary phase (SIII) since they correspond to well-known stages of growth during which bacteria respond to nutrient depletion and accumulate antibiotics.

During submerged growth, we also noticed that *S. rimosus* pellets fragmented and produced a more homogeneous culture than observed for *S. coelicolor* (36). Since the relationship between differentiation processes and secondary metabolite production in submerged *S. coelicolor* cultures has been reported, we used confocal laser scanning microscopy to examine mycelia development (37) and the increase of the antibiotic titer in the culture broth. Various types of mycelia that characterize selected stages of growth are shown in Fig. 1B. From the SI to the SII phase, dispersed hyphae and pellets of increasing size were formed with the simultaneous occurrence of cell death (Fig. 1B, panels a through e). Until the SI phase, the size of the pellets was, on average, 106 $\mu$m $\pm$ 40 ($n$ = 50), while in the SII phase, a few larger pellets (up to ~200 $\mu$m) were observed exhibiting pronounced cell death processes. However, an abundant accumulation of shorter fragments was found, most likely as a consequence of pellet disintegration (Fig. 1B, panels f and g). Interestingly, during the SIII phase, these viable fragments formed tight clumps (Fig. 1B, panels h and i). This could be a consequence of the greater hydrophobicity of the hyphal fragments and/or a result of the extracellular matrix produced at the late stage of growth. Of note, the observed *S. rimosus* mycelium fragmentation was not comparable to mycelial differentiation reported for *S. coelicolor* (5). Although pellet fragmentation is an extremely interesting developmental process, it is outside the scope of this study and remains to be examined in the future. Our goal was to collect biomass during specific growth phases in which a metabolic switch, important for antibiotic production, occurs. Production of OTC was first detected during the SI phase whereby a metabolic switch triggers antibiotic production during growth arrest in *S. coelicolor* (5, 34, 35). The antibiotic titer increased during *S. rimosus* growth and finally reached a titer of 7.04 $\mu$g/mL in the SIII phase (Fig. 1A). As we used wild-type strain G7 and a medium with a high concentration of glucose (Fig. 1A), which was standardized for low and high producers, this could have an impact on antibiotic biosynthesis, but note that OTC was still produced in detectable amounts, indicating that cellular processes important for production of secondary metabolites are switched on. Altogether, these experiments defined the framework to investigate the (phospho)proteome at the critical points of growth and OTC production.

**A significant number of (phospho)proteins from various functional categories were identified during submerged growth of *S. rimosus*.** To determine protein phosphorylation concomitant to the increase in OTC production and pellet fragmentation, we analyzed the (phospho)proteome of *S. rimosus* at three specific stages of growth, described above. We performed stable isotope dimethyl labeling of peptides obtained from proteins at the three different stages of growth. Mass spectrometry (MS) measurements identified 200,982 MS/MS spectra corresponding to 33,023 peptides and 3,725 protein groups (45.6%

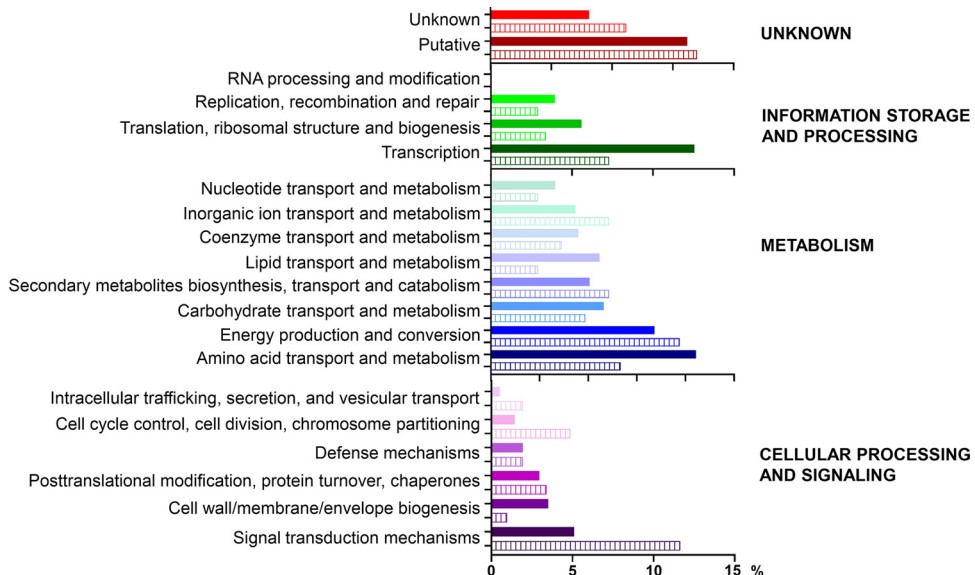

**FIG 2** Distribution of (phospho)proteins into functional subcategories. Percentages of various subcategories obtained for *S. rimosus* proteome (3,448 proteins) and phosphoproteome (206 phosphoproteins) are shown by solid (proteome) and dashed bars (phosphoproteome). Note that only proteins assigned to one functional subcategory were used for this analysis (i.e., 93% of the proteome and, i.e., 90% of the phosphoproteome).

of the theoretical *S. rimosus* proteome) (see sheet 1 of Data Set S1 in the supplemental material). This was in agreement with a previous analysis of *S. coelicolor* grown on solid medium (17). To evaluate reproducibility between the two biological replicates, we calculated the correlation coefficients for each protein ratio. SIII/SII and SIII/SI showed a high correlation between replicates (0.82 and 0.84, respectively), whereas the correlation for SII/SI was somewhat lower (0.64). In the phosphoproteome analysis, 230 phosphoproteins containing 417 nonredundant phosphorylation sites were detected (Data Set S1, sheet 2). Of these, we were able to localize 315 phosphorylation sites to a specific amino acid residue with high confidence (localization probability $\geq$ 0.75). At the phosphoproteome level, correlation coefficients between replicates were 0.55 for SII/SI phosphopeptide ratios, 0.75 for SIII/SI, and 0.74 for SIII/SII.

The distribution of Ser/Thr/Tyr phosphorylation was in line with previous observations (15, 16), with threonine phosphorylation predominant (53.48%), followed by serine phosphorylation (41.25%) and then tyrosine phosphorylation (5.28%). The prevalence of phosphothreonine we attribute to different amino acid frequencies in the *S. rimosus* proteome, where T is more frequent than S and Y.

Our major goal was to investigate whether there were any functional subcategories/categories among the identified proteins that were prevalent in the phosphoproteome in relation to the proteome. Functional annotation of all identified (phospho) proteins was performed using eggNOG-mapper (Data Set S1; sheets 1 and 2). Their distribution into 20 functional subcategories is shown in Fig. 2, while distribution in major categories is shown in Fig. S1.

The most prevalent category in both the proteome and phosphoproteome was "metabolism" (40% and 35%, respectively), followed by "unknown" (23% and 27%, respectively). Although proteins with unknown function remain to be investigated, their abundance suggests important biological roles at various stages of bacterial growth, while their presence in the phosphoproteome suggests that their function may be regulated by phosphorylation. On the other hand, the prevalence of proteins that belong to "metabolism" in the proteome and phosphoproteome is not surprising, as many proteins that participate in primary and secondary metabolism are found to be phosphorylated in diverse bacterial species (38). A more detailed analysis of subcategories belonging to "metabolism" showed a higher prevalence of proteins from "lipid

transport and metabolism" and "amino acid transport and metabolism" in the proteome than in the phosphoproteome. Similar results were obtained for three subcategories in "information storage and processing." In contrast, three subcategories from the category "cellular processes and signaling" were increased among phosphoproteins (Fig. 2). A detailed inspection of these proteins revealed a high number of various regulatory proteins (e.g., Ser/Thr kinases, PTPA2 phosphatase, His-kinase, anti-sigma factors, stress response, etc.) and proteins involved in cell division (e.g., ParA, FtsZ, SepF, DivIVA, etc.). Altogether, the presented results indicate that phosphorylation as a regulatory mechanism is employed in all functional subcategories. However, these findings also support the critical role of phosphorylation in signaling pathways regulating diverse cellular processes, including those that control cell division (38).

To gain an insight into some global trends for protein phosphorylation in other streptomycetes, we compared our results (Fig. S1A) with (phospho)proteomes of *S. coelicolor* (16, 17). In both studies, approximately 45% of the proteomes were identified. We subjected *S. coelicolor* proteins to the same functional categorization as for *S. rimosus* proteins and obtained comparable results in the proportion of functional categories of both proteomes (Fig. S1B). However, for the identified phosphoproteins, the trend in the functional distribution was somewhat different in *S. coelicolor*, where the proportion of the "information storage and processes" category was significantly increased (23%) (Fig. S1B). This could be ascribed to the fact that *S. coelicolor* was grown in the solid medium, as it is well-known that different growth conditions can cause drastic changes in phosphoproteomes, even in the same species (38).

**Quantitative (phospho)proteomics analysis uncovers growth-dependent proteins during morphological changes and antibiotic production in *S. rimosus*.** Our next goal was to gain a detailed insight into proteome and phosphoproteome dynamics and to test any correlation with observed morphological differentiation and antibiotic production during bacterial growth.

**(i) Proteome dynamics.** All proteins that had at least one measured protein ratio between two growth phases in at least one of the two biological replicates (3,353/3,725) are listed in Data Set S1 (sheet 3). To identify statistically relevant proteins that change in abundance during growth, we performed a $t$ test ($P < 0.05$ and $s^0 < 0.1$). This analysis revealed 494 proteins with significant changes in at least one protein ratio (Data Set S1, sheet 4). The distribution of proteins with significant changes in their abundances is shown for three ratios in Fig. 3.

Next, we performed hierarchical clustering to identify proteins with similar temporal abundance profiles (Fig. 4A; Data Set S1, sheet 4). Proteins were clustered in the heatmap (Fig. S2), and the number of clusters was fixed at five in order to gain insight into protein groups that are most represented in any of the selected stages of bacterial growth.

For each cluster, we show (Fig. 4B) the most relevant enriched terms (22 in total) that provide a detailed insight into protein biological functions. Finally, we describe some of the upregulated proteins from the presented clusters, which, according to their functions, are likely to have a prominent role during the specific growth phase in which their increase occurred.

**(a) Proteins upregulated in the SI phase.** Although cluster A did not show any enriched annotation terms, as it comprised only 16 proteins which are distributed in various subcategories (Fig. S3), all of them participate in different cellular processes that are essential for basic metabolism and growth, which correlates with the physiological state of the SI phase. Nevertheless, proteins which were found in the greatest abundances during the SI phase exhibited a cluster B profile. These proteins are needed for vital cellular processes, such as proteins involved in transcription, cell wall/membrane biosynthesis, and others (Fig. 4B; Data Set S1, sheet 4). We describe some of these which either contribute to exponential growth or likely participate in growth arrest. For example, two proteins involved in the synthesis of DNA (NrdA and NrdB) were previously reported to be expressed in early exponential growth in bacteria (39). This result is not surprising due to the presence of many observed live hyphal

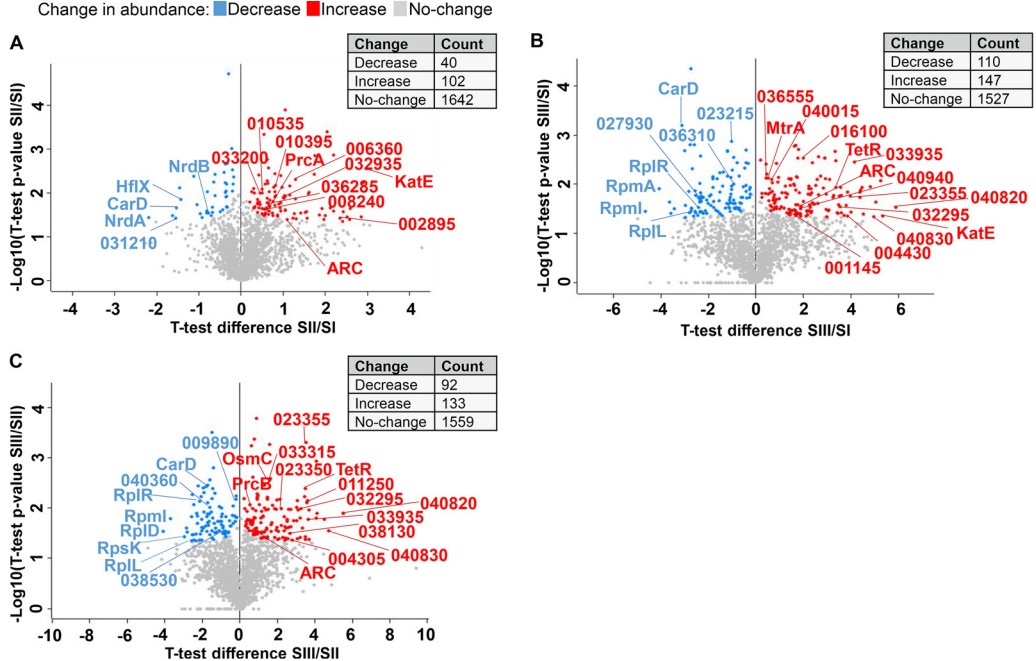

**FIG 3** Distribution of differentially expressed proteins during *S. rimosus* growth. The *y* axis ($-\log_{10}$ *P* value) represents the level of significance of each protein, while the *x* axis ($\log_2$ fold change) represents the difference in protein ratios between the two different growth phases derived from two replicates (A, SII/SI; B, SIII/SII; C, SIII/SII). Significantly upregulated proteins in a given ratio are represented by red circles, and downregulated proteins are represented by blue circles. Proteins described in detail below are marker with corresponding name (if possible) or with the number of SRIM locus tag.

fragments in SI (Fig. 1B). Consistent with this, we also found some ribosomal proteins (RpmI, RpmIL, and RpsK) which were upregulated throughout the exponential phase of growth (SI to SII), while their significant decrease was observed in the late stationary phase, as reported previously (17, 40). Besides these, we also detected proteins related to stress, such as HflX, ribosome-associated GTPase, and a CarD family transcriptional regulator. The protein HflX is involved in bacterial stringent response, as it prevents ribosome biogenesis (41), whereas CarD regulates transcription. In response to various stresses, levels of CarD are highly upregulated (~3- to 20-fold) in *Mycobacterium tuberculosis* (42). Upregulated stress proteins in the SI phase most likely reflect regulatory mechanisms which stop bacterial growth possibly triggered by pellets exhibiting a cell death process (Fig. 1B, panels b through e).

**(b) Proteins upregulated in the SII phase.** Although many proteins that belong to cluster B were constantly upregulated from the SI to SII phases, the most interesting are proteins belonging to cluster C that were specifically upregulated in the SII phase (Fig. 4A; Data Set S1, sheet 4). This cluster is significantly enriched in proteins related to secondary and lipid metabolism, which are important for antibiotic production that increased during this phase. This is also in line with the proteins detected during later phases of growth in *S. coelicolor* (17). Thus, enrichment of these two subcategories during the SII phase is unsurprising. Moreover, Manteca et al. (5) reported that cell death and changes in mycelial morphology correlate with the production of specialized metabolites. We also observed pronounced cell death processes, but mycelial morphology during this phase (i.e., pellet disintegration) significantly differed from that reported for *S. coelicolor*. However, in both cases, these morphological changes coincided with the detection of proteins important for antibiotic synthesis (Fig. 1B). Overall, out of 29 differentially upregulated proteins which belong to secondary metabolism, 11 proteins were upregulated

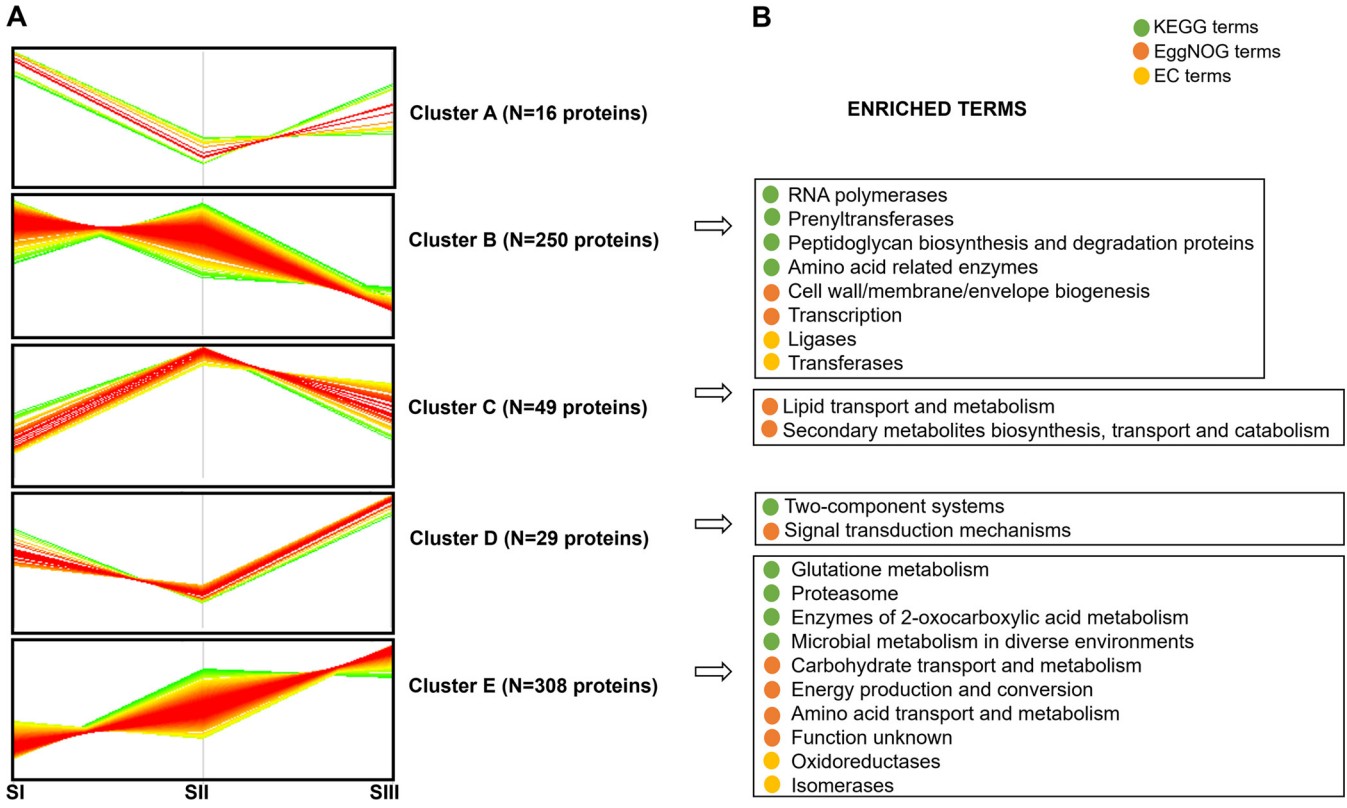

**FIG 4** Clusters of proteins with similar temporal abundance profiles and their functional enrichment analysis. (A) Proteins with significant changes in their abundances determined by *t* test ($P < 0.05$) during the *S. rimosus* growth (SI, SII, and SIII) are grouped into five clusters according to their Z-scored normalized log$_2$-transformed proteins ratios. (B) Functional annotation terms from KEGG, eggNOG, and EC are enriched in different clusters using Fisher's exact test (see Table S1 in the supplemental material).

only in the SII phase (Fig. 4A; Data Set S1, sheet 4) highlighting their importance for elevated antibiotic production.

**(c) Proteins upregulated in the SIII phase.** Interestingly, the highest number (337) of upregulated proteins was found in the SIII phase, belonging predominantly to clusters D and E. Cluster D was significantly enriched in "signal transduction mechanisms" and "two-component systems" (Fig. 4B). The majority of these proteins participate in the stress response (e.g., MtrA, OsmC, and KatE); thus, their detection in SIII implicates a protective role to help the bacterium survive harsh conditions. In addition, 15 proteins involved in secondary metabolism were also upregulated in SIII (Data Set S1, sheet 4), highlighting that their activities are also very intensive during later stages of growth in which antibiotic accumulation occurs. Finally, cluster E was significantly enriched for proteins of unknown function and proteins participating in different primary metabolism processes (e.g., energy production and conversion and amino acid and carbohydrates metabolism). Rioseras et al. (17) reported a similar trend for *S. coelicolor*, suggesting that increased levels of these proteins contribute to the energy and precursors necessary for secondary metabolism in the later stages of *Streptomyces* growth. As expected, this cluster was enriched for proteins associated with the proteasome (e.g., ARC, PrcA, PrcB), most probably needed in higher levels for the maintenance of minimal protein synthesis required for survival of bacteria in stress conditions in SIII (43). We also observed an enrichment of glutathione metabolism, which serves to protect the cell from oxidative and osmotic stresses (44). The transcriptional regulator from the TetR family (SRIM_040805) was also significantly upregulated, which might reflect a possible role in accumulation of OTC in the medium. Namely, it is known that the members of this family control the expression of the *tet* genes, whose products confer resistance to tetracycline (45).

In summary, the proteome dynamics of *S. rimosus* presented here define, for the first time, protein groups, i.e., proteins upregulated at selected growth stages, thus reflecting their importance during very specific physiological states in which growth arrest, antibiotic production, and pellet fragmentation occur. Noteworthy, OTC biosynthesis was at low levels, and only 13 out of 25 of the proteins involved in the biosynthetic pathway were detected in the proteome, but not at a statistically significant level (Data Set 1, sheet 1). However, consistent with the onset of OTC synthesis, they first appeared during the SI phase, indicating that the bacterium entered a complex developmental program involving cell death, morphogenesis, and antibiotic production. In addition, if we apply an arbitrary cutoff of 1.5 on a $\log_2$ scale (see below quantification of phosphoproteome), then most of the detected proteins showed significant upregulation during the SII and SIII phases following an increase in OTC in the medium (Data Set 1, sheet 3).

**(ii) Phosphoproteome dynamics.** As for the proteome, we examined the dynamics of protein phosphorylation during the SI, SII, and SIII growth stages. Out of 417 phosphorylation sites identified in this study (Data Set S1, sheet 2), 191 were quantified in at least one replicate (Data Set S1, sheet 5). To identify phosphorylation sites with changes in abundance, we defined an arbitrary threshold of a 3-fold change (1.5 on a $\log_2$ scale) for phosphorylation site ratios. With this approach, we identified 71 proteins exhibiting 98 differentially regulated phosphorylation sites (Data Set S1, sheet 5). Among them, 55 proteins had only one, while 16 proteins had multiple upregulated phosphorylation sites. These phosphoproteins, grouped into six major functional categories as described in Materials and Methods, were divided between growth phases in accordance with the upregulation of their phosphorylation site(s) (Data Set S1, sheet 6). The number of phosphoproteins/phosphorylation sites (unique and total) was increased in each phase (Fig. 5A), while the numbers of phosphoproteins and phosphorylation sites that are shared between growth phases are shown in Fig. 5B and C, respectively. The highest number of phosphoproteins/phosphorylation sites are upregulated in the SII phase, during which bacteria sense nutrient depletion and slow their growth, while most of the shared phosphoproteins/phosphorylation sites are observed in the SI and SII phases. This may be attributed to the heterogeneity of mycelium, which was more pronounced in the SI and SII phases than in the SIII phase. We describe some phosphoproteins according to the temporal abundance profiles of their phosphorylation sites found uniquely during selected growth phases and also highlight those that have not been found in any prokaryote so far (Fig. 5D). The identification of this posttranslational modification (PTM) might be relevant for the specific physiological state observed in that certain phase and could provide useful information in future studies.

**(a) Protein phosphorylation in the SI phase.** During the SI phase, 4/5 phosphoproteins (Fig. 5De) belong to the category of "regulatory proteins," including Ser/Thr protein kinase (SRIM_020800), phosphatase with dual-substrate specificity (SRIM_021470), transcriptional regulator (SRIM_021055), and ribosome silencing factor RsfS (SRIM_013715), while the protein SRIM_040820 belongs to the category "secondary metabolism" (a member of the SDR family oxidoreductase). These proteins most likely have a significant role in triggering antibiotic production and in the observed cessation of cell proliferation (5). For example, it is known that ribosomal silencing factor (RsfS) slows bacterial growth by inhibiting protein synthesis (46), while the transcriptional regulator SRIM_021055 belongs to the xenobiotic response family of transcriptional regulators. One member of this family was reported to control antibiotic production and development in *S. coelicolor* (47). Although we describe here only phosphoproteins that were uniquely phosphorylated in SI, we also highlight SRIM_040830 found in SII since it belongs to the category "secondary metabolism." Notably, SRIM_040830 and SRIM_040820 do not have orthologs in *S. coelicolor*; thus, they could have an important and specific role in antibiotic production in *S. rimosus*.

**(b) Protein phosphorylation in the SII phase.** Most of the phosphoproteins identified in the SII phase belong to the category "regulatory proteins" and include three STPKs (SRIM_018190, SRIM_020795, and SRIM_025105), one Ser/Thr protein phosphatase (SRIM_020985), one sensor His kinase (SRIM_027850), one FadR transcriptional

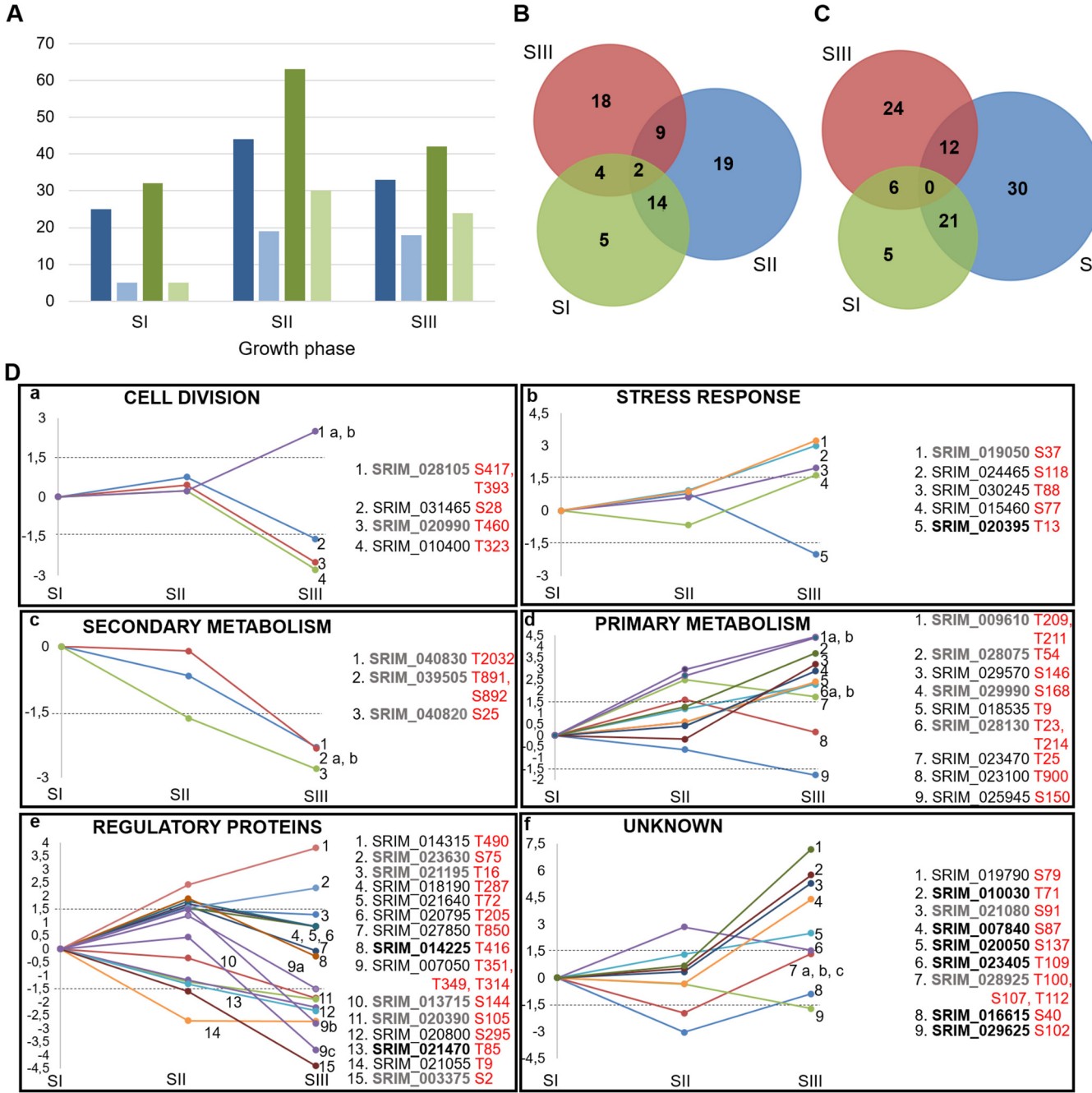

**FIG 5** Distribution and dynamics of proteins with upregulated phosphorylation sites during *S. rimosus* growth. (A) Number of all and unique upregulated phosphoproteins/phosphorylation sites by growth stages (dark blue, all proteins; light blue, unique proteins; dark green, all phosphorylation sites; light green, unique phosphorylation sites) (B and C) Venn diagrams represent phosphoproteins (B) and phosphorylation sites (C) shared between growth phases. (D) Dynamics of proteins with upregulated phosphorylated sites throughout growth phases (the *y* axis shows log$_2$ transformed ratios of phosphorylation sites in each growth stage shown on the *x* axis; phosphoproteins marked in bold are reported here for the first time, streptomycetes are in gray, and any other prokaryote are in black).

regulator (SRIM_021640), and one VWA-containing protein (SRIM_014225). The prevalence of regulatory proteins reflects the importance of signaling networks for bacterial adaption to starvation, which employs phosphorylation to regulate cellular processes during the entry into the stationary phase. Also, other categories ("stress response," "primary metabolism," and "cell division") were represented by a smaller number of proteins. For example, we identified two phosphorylated stress response proteins with a TerD domain (SRIM_012680 and SRIM_020400) shown to be important for redox stress adaptation (48) and two phosphorylated proteins in the category of "primary

metabolism" (DNA-directed RNA polymerase subunit beta RpoB, SRIM_018645, and thymidylate kinase, dTMP, SRIM_023100). For dTMP kinase, which is crucial for DNA replication and repair, we found three upregulated phosphorylation sites, which likely indicates the importance of this PTM for the enzyme activity during the late exponential stage of growth. We also identified upregulated phosphorylation sites for three proteins belonging to "cell division" (SRIM_014620, SRIM_015590, and SRIM_026815), which were reported to be phosphorylated in actinobacteria.

**(c) Protein phosphorylation in the SIII phase.** Interestingly, in the SIII phase, most of the upregulated phosphorylation sites belong to proteins from "unknown" (6), "primary metabolism" (6), and "stress" (5) categories (Data Set S1, sheet 6). The increased phosphorylation of proteins likely participates in the regulation of the processes allowing bacteria to adapt to stress and starvation. For example, phosphorylation of ribosomal proteins was reported for many distantly related bacteria (28). Here, we report three ribosomal proteins with upregulated phosphorylation sites (30S ribosomal proteins S16, SRIM_029570, and S17, SRIM_018535, and 50S ribosomal protein L31, SRIM_028075). This PTM most likely participates in attenuating protein synthesis during the SIII phase, as it was reported that ribosomal protein phosphorylation affects protein translation and decreases protein synthesis in *Streptomyces* (49). Interestingly, during proteome quantification analysis, we detected four additional ribosomal proteins, which were significantly decreased in SIII (see previous section). Altogether, these examples show how bacteria employ two mechanisms to slow protein translation, either by decreasing the concentration of ribosomal proteins or by inhibiting their function via phosphorylation. Regarding stress response proteins, we identified phosphorylated Dps protein (DNA-protecting protein under starved conditions, SRIM_030245), which is induced by oxidative or nutritional stress, as well as cochaperone GroES (SRIM_018295). Members of these protein families are found to be phosphorylated across the bacterial kingdom, strongly suggesting that they could be functionally regulated by phosphorylation (28, 50). In addition, we identified one stress response TerD family protein (SRIM_024465); a cold shock domain-containing protein (SRIM_019050), which might participate in oxidative stress tolerance (51); and a DNA-binding protein Hu (SRIM_015460), which can protect the chromosome in response to stress (52). Strikingly, proteins belonging to the "unknown" category exhibited the highest levels of differential phosphorylation (SRIM_019790, SRIM_010030, SRIM_021080, and SRIM_007840; up to 128-fold change) (Fig. 5D, panel f) in the SIII phase, likely suggesting that their phosphorylation has a strong impact on regulating the biological roles of these proteins.

Altogether, during the SI and SII phases, we found mostly regulatory proteins with upregulated phosphorylation sites, indicating their prominent role in complex and dynamic signaling networks associated with growth arrest and metabolic switch, which leads to antibiotic production. During the SIII phase, most of the identified proteins have predicted functions essential for bacterial survival and adaptation to harsh conditions, which indeed reflects the limitation of essential nutrients and the presence of various stress factors. Accordingly, the detected upregulation of numerous uncharacterized proteins at the (phospho)proteome level strongly suggests that their biological role is connected to physical and chemical stresses.

**Comparison of *S. rimosus* and *S. coelicolor* phosphoproteomes reveals a low number of conserved phosphorylation sites among orthologs.** Comparison of *S. rimosus* and *S. coelicolor* phosphoproteomes can point to specific regulatory pathways crucial to these complex prokaryotes (e.g., antibiotic production or mycelium differentiation) or some global cellular trends. In this context, we compared *S. rimosus* phosphoproteome with the 170 phosphoproteins found in *S. coelicolor* (15–17). Altogether, Parker et al. (15) identified 40 phosphoproteins from bacteria grown in liquid medium during the mid-exponential phase using preparative isoelectric focusing for protein fractionation. A significantly higher number of phosphoproteins were identified, $n = 127$ (16), and $n = 85$ (17), respectively, with the global gel-free liquid chromatography-tandem mass spectrometry (LC-MS/MS) approach from *S. coelicolor* grown on solid media. We compared *S. rimosus* and *S. coelicolor* phosphoproteomes using

Diamond software (53) and found only 47 phosphorylated orthologs, while 28 additional were found in the dbPSP (54). Altogether, of *S. rimosus* 230 phosphoproteins, 155 (67%) were not found among previously reported *S. coelicolor* phosphoproteins. The majority of these proteins belong to the categories "metabolism" and "unknown," suggesting that phosphoproteomes in bacteria are still very much understudied. In addition, we provide information for those found to be differentially phosphorylated (Fig. 5D), which can help in future research by elucidation of their biological role during very specific growth phases. Subsequently, we aligned 47 phosphoproteins of *S. rimosus* with their matching *S. coelicolor* orthologs obtained by Diamond software. Based on the conservation of their phosphopeptides/phosphorylation sites, we generated three categories, (i) phosphopeptides with highly conserved phosphorylation sites, (ii) overlapping phosphopeptides with different phosphorylation sites, and (iii) nonconserved phosphopeptides (Table 1 and Tables S3 and S4). Identical or highly conserved phosphopeptide/phosphorylation sites are found mostly among proteins that have various regulatory functions, such as kinases, anti-sigma factor antagonists, and transcriptional regulators (Table 1). Interestingly, orthologs with overlapping phosphopeptides and different phosphorylation sites generated the smallest group of 10 phosphoproteins (Table S3). However, most of these proteins also exhibit various regulatory functions. All other phosphorylated orthologs from *S. rimosus* and *S. coelicolor* that did not exhibit evolutionary conserved phosphopeptides were listed (Table S4), showing that these proteins exhibit a wider range of biological functions. Altogether, among *S. rimosus* phosphoproteins identified here, only 20.4% were found phosphorylated in *S. coelicolor*. Nevertheless, 43% of these orthologs share highly conserved phosphorylation site(s), suggesting that this PTM is functionally conserved between the two species.

**Phylogenetic analysis of *S. rimosus* STPKs confirms relatedness to *S. coelicolor* STPKs and indicates their biological roles.** The genome of *S. rimosus* predicted the presence of 32 STPKs, while 47 STPKs were predicted for *S. coelicolor* (16). Since closely related proteins often share similar biological functions (55), we constructed a phylogenetic tree using sequences of STPKs from *S. rimosus* and *S. coelicolor* that were identified in various (phospho)proteomic studies. We expected that phylogenetic clustering, together with the information (if available) about the growth point in which some kinase has been identified, or the dynamics of its phosphorylation might provide a clue about its biological role and create directions for future research. Despite the similar coverages of identified kinases in both organisms (~57%), our study detected 12 phosphorylated kinases in *S. rimosus*, while only 4 were found in *S. coelicolor* (15, 16). Since kinases SCO3821, SCO4423, and SCO1551 were detected while growing *S. coelicolor* in submerged culture (15) and only SCO2666 was detected while the bacterium was growing on solid medium, we speculate that their activities depend on growth conditions. With respect to phylogenetic relationships, 15 out of 18 STPKs detected in the *S. rimosus* (Fig. 6) (phospho)proteome clustered together with previously detected *S. coelicolor* kinases. For comparison, we also added to Fig. 6 the phosphorylation status of each kinase and the growth phase in which we identified an upregulation (Table S2). For example, *S. rimosus* kinase (SRIM_020800), found to be upregulated uniquely in the SI phase, closely relates to a membrane *S. coelicolor* PksC kinase (SCO3821). This membrane-bound kinase possesses a PASTA motif, which binds peptidoglycan fragments and is speculated to participate in cell wall synthesis and cell morphogenesis (56). It was also found in previous phosphoproteomic studies during the exponential phase of growth, suggesting that its biological role is more important during the early stage of bacterial growth. However, its phosphorylation was reported only when the bacteria were grown in liquid culture (15), which also suggests that this kinase is more active during submerged growth. Another interesting example is AfsK, one of the best-studied STPKs in *S. coelicolor*. It controls antibiotic production (57) and regulates polar growth, hyphal branching (58), and initiation of chromosome replication in *S. coelicolor* (59). The *S. rimosus* kinase (SRIM_017990), which grouped with AfsK, was also found phosphorylated in all stages of growth, during which time the bacterium continuously synthesized OTC. Thus, our data strongly suggest

**TABLE 1** Phosphopeptides with highly conserved phosphorylation sites between *S. rimosus* and *S. coelicolor*[a]

| Gene ID | Phosphopeptides alignment | Predicted function | % Sim. | Ref. |
|---|---|---|---|---|
| SRIM_026575 | `320AAEESITFSLPK331`<br>`   ||||||.|||||` | 4-hydroxy-3-methylbut-2-enyl diphosphate reductase | 84.1 | this study |
| SCO5058 | `302AAEESIIFSLPK313` | hypothetical protein | | (16) |
| SRIM_023470 | `1-----MTSSDPTVQPVPTSVVERVDAADVTLSNPKR31`<br>`     ...|||||||||..||:|:||||||.|||||||` | 2-amino-4-hydroxy-6-hydroxymethyldihydropteridine diphosphokinase | 81.4 | this study |
| SCO3401 | `2SAPFAQGPSDPTVQPVPASVIEQVDAADTTLSNPKR36` | putative hydroxymethyldihydropteridine pyrophosphokinase | | (16, 17) |
| SRIM_025945 | `138HLGAVAGVEVTASHNPPRDNGYK160`<br>`   ..|...||..||.|||||.|.||.` | phospho-sugar mutase | 27.5 | this study |
| SCO7443 | `134TSGLADGVVVTPSHNPPADGGFK156` | phosphoglucomutase | | (16, 17) |
| SRIM_018295 | `17IVVQPLDAEQTTASGLVIPDTAK39`<br>`  |||||||||||||||||||||||` | co-chaperone GroES | 98 | this study |
| SCO4761 | `17IVVQPLDAEQTTASGLVIPDTAK39` | 10 kD chaperonin cpn10 | | (15–17) |
| SRIM_024360 | `150AEALLDEVLAS160`<br>`   |||||||||||` | CarD family transcriptional regulator | 100 | this study |
| SCO4232 | `150AEALLDEVLAS160` | putative transcriptional factor regulator | | (17) |
| SRIM_020800 | `175AMQSGVTSMTQTGMVVGTPQYLSPEQALGR204`<br>`   ||||||||||||||||||||||||||||||` | protein kinase | 61.6 | this study |
| SCO3821[b] | `175AMQSGVTSMTQTGMVVGTPQYLSPEQALGR204` | serine/threonine protein kinase | | (15) |
| SRIM_023100[c] | `924SQPSSASSDAEETAVLPQHRDVPGASDETAVLPPVRPEAR963`<br>`   ...:|..|:||..||:.....:||||||||.|:|..` | thymidylate kinase | 75 | this study |
| | `951GGRASGESESEVTTELPKPPVPSGAADETAVLPAVQPRDA990` | | | |
| SCO3542 | `990EEGGRTDRRDRTDARGDSGTSGVERTRELPQIDPATGEPVEPAPR1029`<br>`   :|.. ..:||||||||| . |: |`<br>`1015EESPAE-------------EAQDRTRELPQIDPDQA---PPSRR1042` | integral membrane protein with kinase activity | | (16, 17) |
| SRIM_001820 | `45DRARGVLIDISRLEIIDSFVAR66`<br>`  ..|.||:|:|.:||:|||:.|` | STAS domain-containing protein | 48.3 | this study |
| SCO7323 | `53SGATGVVIDVSGVEIVDSFLGR74` | anti-sigma factor antagonist | | (17) |
| SRIM_007050 | `346--AANTPTP------RPQAPR358`<br>`   ...||.| |..|||` | protein kinase | 62.6 | this study |
| SCO1551 | `324PGPQGTPAPRGTSASRGSAPR344` | putative eukaryotic-type protein kinase | | (15) |
| SRIM_023050 | `50ERVDFLDSTGLGVLVGGLK68`<br>`  |.|||||||||||||||||` | anti-sigma factor antagonist BldG | 97.3 | this study |
| SCO3549 | `50EGVDFLDSTGLGVLVGGLK68` | putative anti-sigma factor antagonist | | (16) |
| | `49ERVDFLDSTGLGVLVGGLK68`<br>`  .|::|.|||||.||:|...` | putative anti anti sigma factor | | |
| SCO3067 | `61SRLEFCDSTGLNVLLGARL79` | | | |
| SRIM_014215 | `120SLSGTST-RSMPVSVR134`<br>`   ::||.|| || ||||` | protein kinase | 67.3 | this study |
| SCO2666 | `100AVSGRSTGRS--VSVR113` | putative Ser/Thr-protein kinase | | (16) |
| SRIM_010385 | `197IAEGGFFNQS206`<br>`   ||||||||||` | Cell division protein SepF | 79.3 | this study |
| SCO2079 | `204IAEGGFFNQS213` | conserved hypothetical protein | | (16, 17) |
| SRIM_010375 | `274QLENQADDSLAPPR287`<br>`   |||.|||||||||` | DivIVA domain-containing protein | 67.6 | this study |
| SCO2077 | `301QLETQADDSLAPPR314` | hypothetical protein | | (16, 17) |
| SRIM_024125 | `38LLDSTGEFTAEVPTSATGQFR58`<br>`  |||:||||||||||||||||` | DUF1416 domain-containing protein | 91.6 | this study |
| SCO4165 | `38LLDATGEFTAEVPTSATGQFR58` | conserved hypothetical protein | | (17) |
| SRIM_014225 | `404AVQLASASGNEDTAK418`<br>`   |||||..|||.||||` | VWA domain-containing protein | 80.9 | this study |
| SCO2668 | `396AVQLAGVSGNADTAK410` | conserved hypothetical protein | | (15–17) |
| SRIM_024700 | `281VDPLPLRPDRAGGS--VEESGPGEEFGHS307`<br>`   |||.||||...:| |.|.....|.|||` | DUF3027 domain-containing protein | 73.4 | this study |
| SCO4330 | `280VDPFPLRPAADSGSVPVTEDQAVAELGHS308` | conserved hypothetical protein | | (16) |
| SRIM_021055 | `103TAGGSVLNNTAQTATSTGATTVRR126`<br>`   :.||||||:||..|.:|:||.||:|` | DNA-binding protein | 83.5 | this study |
| SCO3859 | `103SQGGSVLSNTTTTTSSSGAPTVKR126` | putative DNA-binding protein | | (16, 17) |
| SRIM_016410 | `187ADEGGDDRSVGLPETVFAPPLAGFDEQDTPPPSTRPEAK225`<br>`   ..:.|||||||.||.:|||||||:|.|: ||||||...:||` | hypothetical protein | 61.1 | this study |
| SCO3115 | `218-ADAGDDRSVPLPQTVFAPPLSGVDD-DTPPPSTTSDAK254` | hypothetical protein | | (15–17) |
| SRIM_003375 | `2SPGQTPPGPHPVDDGLVR19`<br>` |.|||.|....|||.|||` | aminoglycoside phosphotransferase family protein | 77.7 | this study |
| SCO7478 | `2SSGQTHPDRCRVDDRLVR19` | putative phosphotransferase | | (15) |
| SRIM_028800 | `242DAYAAGPAPAQAPAPAYGYGYPRPY----266`<br>`   :|:| |||||||| ||||| |` | enhanced serine sensitivity protein SseB | 73.5 | this study |
| SCO5475 | `242EAHA--PAPAQAPAG--GYGYP-PAGSRY265` | hypothetical protein | | (15) |
| SRIM_026815 | `16GGAAYPSGTPPYGTPREADDAAAADA--41`<br>`  ||||||||||||||||..:| |.|||` | hypothetical protein | 41.2 | this study |
| SCO5128 | `16GGAAYPSGTPPYGTPTASD--AGADAGR41` | putative membrane protein | | (15–17) |

[a]Red letters indicate phosphorylation sites; dark gray color indicates phosphopeptide sequence, and light gray color indicates flanking regions of phoshopeptides.
[b]Ambiguous phoshosites.
[c]Two phoshopeptides in a row are marked with the two shades of gray color.

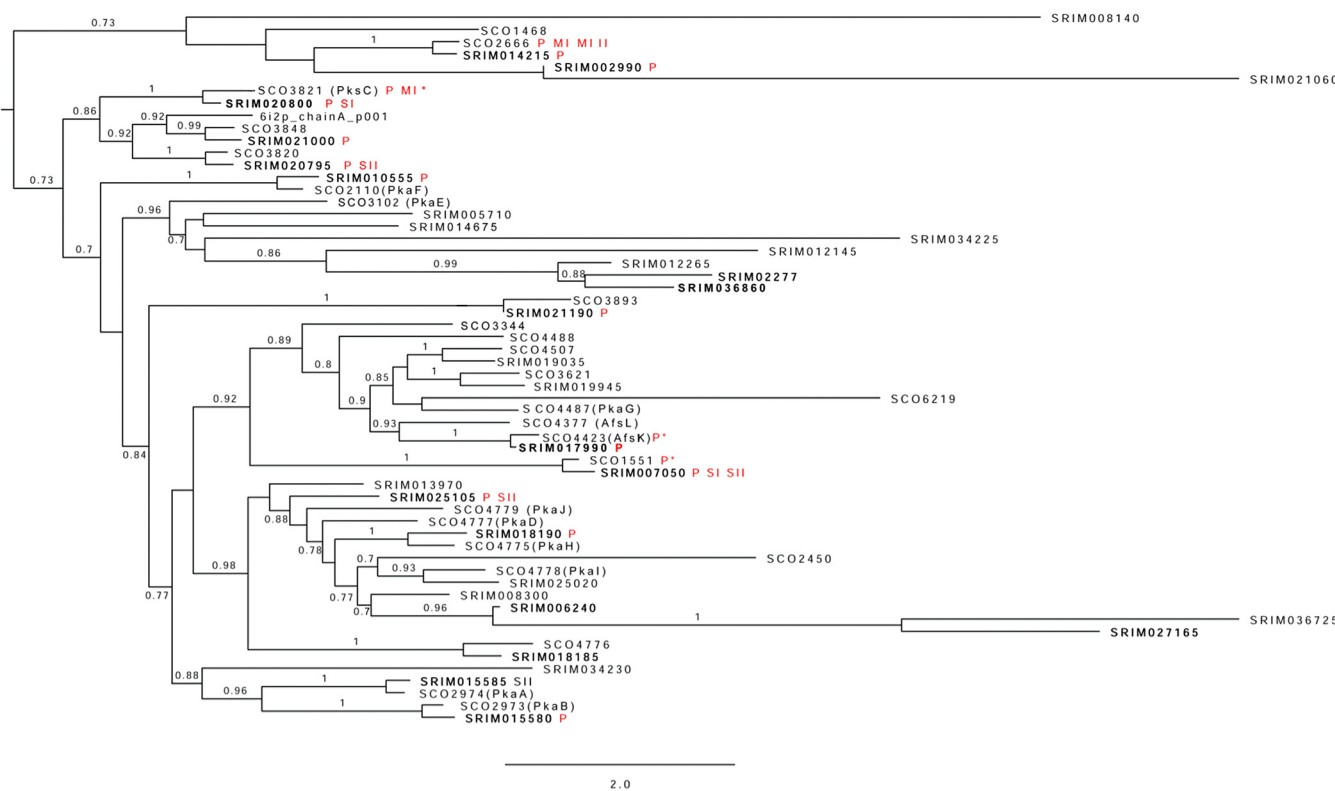

**FIG 6** Phylogenetic tree of STPK sequences from *S. rimosus* G7 and *S. coelicolor* M145. Experimentally detected *S. rimosus* kinases are shown in bold. Phosphorylated kinases were marked by a red P, and information regarding growth phase (SI, SII, or SIII) was added if available. Phosphorylated *S. coelicolor* kinases found by the Manteca group (16, 17) were marked similarly using MI (12 h or 16 h), MII (24 h or 30 h), and MIII (65 h or 72 h) for growth stages. Kinases found by Parker et al. (15) were marked with asterisks. *M. tuberculosis* kinases (PDB IDs 6I2P and 3OUV) were used as outgroups. aLRT values are shown for main branches.

that SRIM_017990 kinase likely has an important role in regulating OTC synthesis. Finally, we point out a cluster with PkaI and PkaH kinases involved in the regulation of sporulation septation and spore wall synthesis (60). We speculate that *S. rimosus* kinases, which exhibit phylogenetic relatedness to these kinases, might be involved in the regulation of pellet fragmentation in submerged cultures of *S. rimosus* when bacteria enter into the stationary phase (61).

**Analysis of *S. rimosus* phospho-orthologs within prokaryotes revealed 55 previously unreported phosphoproteins.** To gain a better insight into the phosphorylation of orthologs within different prokaryotic species, we analyzed identified phosphopeptides using dbPSP. Besides five *S. rimosus* phosphopeptides conserved in *S. coelicolor* and already shown in Table 1, we could not find any other identical phosphopeptides in dbPSP. This result is consistent with the present knowledge that phosphopeptides are not well conserved within proteins from closely related bacteria even if they share the same biological function (14). Next, we submitted sequences of 230 phosphoproteins identified in this study to the dbPSP and found that 175 phosphoproteins were identified previously in various bacteria and archaea (Fig. 7; Table S5A and B). As shown by the Venn diagram, the largest number of *S. rimosus* phosphorylated orthologs (62) shared between bacteria and archaea belong to "metabolism." A similar trend can be seen for "cellular processes and signaling." Note that a high number of phosphoproteins from this category that are found only in Gram-positive bacteria (18) belong to *Actinobacteria*. In contrast, only very few of the *S. rimosus* phosphorylated hypothetical proteins are shared between these analyzed prokaryotic groups. Moreover, for the majority of them (15), their phosphorylated orthologs are found in Gram-positive bacteria, i.e., mostly in *Actinobacteria* (Table S5B). Considering the potential of *Actinobacteria* to produce antibiotics and other bioactive

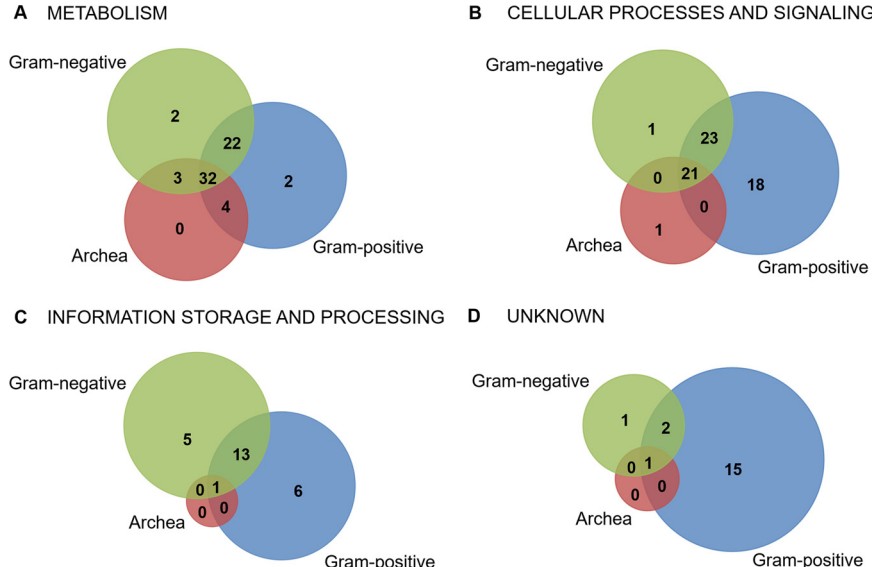

**FIG 7** *S. rimosus* phosphorylated orthologs shared with other Gram-positive and Gram-negative bacteria and archaea.

compounds, phosphorylated orthologs specific to this phylum could be important for production of secondary metabolites and morphological differentiation.

Finally, our study contributes 55 phosphorylated proteins to the database of prokaryotic phosphoproteomes that have not been previously reported (Table S5A). Almost half of these were distributed among "information storage and processes" (1), "cellular processes and signaling" (6), and "metabolism" (9), while 39 remain uncharacterized (Fig. 7; Fig. S6). Eleven of these 55 phosphoproteins exhibited increased phosphorylation at specific growth phases, and the most interesting ones that well represent the physiological state of the bacterium have been described in previous paragraphs (Fig. 5D). Altogether, the dynamics of phosphorylation during specific stages of bacterial growth for proteins unreported prior to this study, in particular for those with unknown function, could direct future investigations and help in elucidating their biological roles.

## MATERIALS AND METHODS

**Bacterial strain, culture conditions, and evaluation of growth.** The wild-type strain *Streptomyces rimosus* subsp. *rimosus* ATCC 10970, also known as *S. rimosus* G7, was used in this study (25). Inocula for all subsequent experiments were obtained from spore suspensions ($10^{10}$/mL) stored in 20% (vol/vol) glycerol at −20°C. Biomass, obtained in the form of short fragments and small mycelial fragments, was collected, homogenized, washed, and a dense suspension prepared in 20% (vol/vol) glycerol solution stored at −20°C for all subsequent cultivation experiments. Precultivation was carried out overnight in 15 mL of seed media (SM; 10 g D-glucose, 5 g yeast extract, 15 g lactalbumin hydrolysate, 2.8 g sucrose, and 1 g CaCO₃ per 1 L) inoculated with 2% (vol/vol) of dense young biomass suspension for subsequent cultivation in TSM5 medium [per 1 L, 50 g glucose, 10 g soya peptone, 7 g MOPS (morpholinepropanesulfonic acid) buffer/free acid, 2 g Ca(NO₃)₂·4H₂O, 2 g NaNO₃, 50 mg NH₄NO₃, 50 mg KCl, 40 mg MgSO₄·7H₂O, and 20 mg ZnSO₄·7H₂O; pH was adjusted to 7.2 by NaOH] optimized for antibiotic production. To obtain good aeration, 300 mL of this medium (in a 2 L flask) was inoculated with a 5% (vol/vol) biomass suspension obtained from SM and incubated at 30°C for up to 120 h. To verify growth reproducibility, bacteria were cultured a total of three times and in two biological replicates. The culture medium was sampled (2 mL) every 2 to 3 h during early to mid-exponential and late exponential to early stationary stages of growth. Once *S. rimosus* reached the stationary phase, when growth ceased, the sampling was decreased to once every 6 h. Bacterial growth was quantified by measuring the dry cell biomass obtained by a vacuum filtration system on a 0.45 $\mu$m membrane filter (Sartorius) using a microwave drying technique (2 min at high power setting).

**Harvesting and cell lysis.** Mycelia for proteomic/phosphoproteomic evaluation were sampled at three selected time points, (i) during exponential phase, during which an arrest of growth can be observed (~15 h of growth); (ii) at late exponential phase when growth slowed and bacteria began to enter the stationary phase (~35 h of growth); and (iii) at late stationary phase (~85 h of growth). The mycelia harvested from 100 mL to 250 mL volume (~500 mg of biomass/sample) were collected by centrifugation at 4,500 rpm for 10 min, washed twice in 10 mL of a lysis buffer (LB; 50 mM Tris-HCl, pH 7.7, 5 mM 2-glycerol phosphate,

5 mM sodium fluoride, 5 mM sodium pyrophosphate, and 1 mM sodium orthovanadate) modified from reference 62 and resuspended in a few milliliters (50 mg/mL) of sonication buffer (LB supplemented with 5 mM sodium tartrate and 5 mM imidazole). Cell lysis was performed by freezing the sample in liquid nitrogen, quickly defrosting, and boiling for 5 min (16). Phenylmethanesulfonyl fluoride (PMSF) was added, after cooling on ice, to a final concentration of 0.5 mM, and the samples were sonicated ($6\times$ 20 s at 40% amplitude) on ice. Benzonase (25 U/mL) and *N*-octylglucoside (1%) were added to the lysate and incubated for 10 min at room temperature as described (62). Cell debris was discarded after centrifugation at 13,000 rpm for 30 min, samples were dialyzed against large volumes of water as described (16), and proteins were precipitated with methanol-chloroform (63).

**Western blot analysis.** Crude protein extracts were analyzed by Coomassie blue staining after running the samples (10 $\mu$g estimated by Bradford assay) on a 12% SDS-PAGE; prior to MS analysis, Western blotting was performed to examine posttranslational modification (PTM), i.e., the occurrence of Ser/Thr/Tyr phosphorylation in the protein samples. Western blot analysis was performed as described previously (26) using anti-phosphoserine/threonine/tyrosine antibodies (Abcam; catalog no. ab15556) for detection of Ser/Thr/Tyr phosphorylation. In addition, Tyr phosphorylation was detected using 4G10 antibodies (Sigma-Aldrich). Antibody binding was visualized with peroxidase-coupled rabbit anti-mouse antibody and chemiluminescent detection (Pierce ECL Western blotting substrate; Thermo Scientific).

**OTC production.** Production of OTC was determined by the agar well diffusion method using OTC-sensitive *Bacillus subtilis* 168 (wild type). OTC production at selected growth phases was quantified by high-pressure liquid chromatography (HPLC). Culture supernatants (900 $\mu$L) were acidified with 0.1 M HCl (100 $\mu$L), incubated for 30 min, and filtered through a 0.45 $\mu$m polytetrafluoroethylene (PTFE) filter. Subsequently, 30 $\mu$L of sample was analyzed using an Agilent 1200 HPLC series system (Agilent Technologies, Germany). Chromatographic separations were performed on a Waters Symmetry C$_{18}$ column (150 mm by 4.6 mm, 5 $\mu$m) using a gradient elution of (A) 0.1% formic acid in water and (B) acetonitrile as follows: 0 to 60 min, 10 to 90% B; 60 to 65 min, 90% B; and 65.1 to 70 min 10% B, at a flow rate of 0.5 mL/min. The column temperature was maintained at 30°C. A standard curve was constructed using various concentrations (1 to 100 $\mu$g/mL) of OTC (Sigma, USA). All samples were monitored at the following wavelengths: 260, 266, 270, 320, 350, 365, and 370 nm, and OTC was detected at 266 nm. The chromatographic data were analyzed using EZChrom Elite software (Agilent Technologies, Germany). The concentration of glucose in samples was determined by ultraperformance liquid chromatography (UPLC) using the Agilent Technologies 1290 Infinity II LC system (Santa Clara, CA, USA) with a Carbo-H$^{++}$ precolumn (4 mm by 3 mm; Phenomenex, Des Plaines, IL, USA), a Rezex ROA column (15 cm by 7.8 mm; Phenomenex), and a refractive index detector. The mobile phase was 0.0025 M H$_2$SO$_4$, the volume of the analyzed sample was 10 $\mu$L, and the flow rate was 0.6 mL/min. The temperature of the column was 60°C. The results were analyzed with OpenLab CDS software.

**Fluorescence microscopy.** *S. rimosus* G7 spores were inoculated into a 100 mL Erlenmeyer flask containing 10 mL seed medium as described, and the culture was shaken at 250 rpm at 30°C. After 24 h, 500 $\mu$L of seed culture was inoculated into 50 mL of TSM5 medium. The culture was sampled at selected time points and stained with propidium iodide (PI) and SYTO 9 green fluorescent nucleic acid stain (Live/Dead Bac-Light bacterial viability kit; Molecular Probes; catalog no. L-13152; Thermo Fisher). In brief, 1 mL of culture was centrifuged and washed with redistilled H$_2$O, and 3 $\mu$L of prepared SYTO 9-PI mixture was added to mycelia following incubation in the dark for 10 min. The samples were observed using a Leica SP8-X FLIM confocal microscope with excitation light wavelengths at 488 nm and 568 nm and emissions at 530 nm (green) or 640 nm (red).

**Protein digestion.** Crude extract preparations were resuspended in a denaturation buffer (10 mM Tris-HCl, pH 8, 6 M urea, 2 M thiourea, and 20 mM ammonium hydrogen carbonate), and protein concentrations were measured by the Bradford method. Protein digestion was performed following the previously described protocol (64). In brief, denatured proteins were reduced using dithiothreitol (DTT) and subsequently alkylated with iodoacetamide (IAA), digested with endoproteinase Lys-C (1:100 [wt/wt]) (Waco), diluted with ammonium bicarbonate, and further digested with sequencing-grade modified trypsin (1:100 [wt/wt]) (Promega). The reaction was stopped by acidifying with trifluoroacetic acid (TFA) to a pH of <3.

**Stable isotope dimethyl labeling.** For quantitative analyses, peptides were loaded onto Sep-Pak C$_{18}$ columns and labeled by dimethylation as described previously (65). The column was activated with methanol and equilibrated with solvent A (0.1% FA). Peptides were loaded and washed twice with solvent A. The column was flushed twice slowly for a minimum of 10 min with freshly prepared labeling reagents prepared as described (66). Labeled peptides were washed twice with solvent A and eluted with 1.8 mL of 80% acetonitrile (ACN) (vol/vol) in 6% TFA (vol/vol). Acetonitrile was evaporated from the eluates by vacuum centrifugation, and the samples were purified using StageTips (see below). The efficiency of labeling was determined by LC-MS/MS measurement of light, intermediate, and heavy labeled samples. Before the main LC-MS/MS proteome analysis and phosphopeptide enrichment, differentially labeled samples were mixed in equal peptide amounts (10 $\mu$g) and subjected to pilot LC-MS/MS measurements for validation of correct mixing.

**Phosphopeptide enrichment.** Phosphopeptides were enriched by TiO$_2$ chromatography where labeled peptides were incubated with TiO$_2$ beads at a ratio of 1:10 (peptide to bead) for 10 min in 8 to 10 consecutive rounds. TiO$_2$ beads were washed as described (67). Phosphopeptides were slowly eluted in three steps, first with 30 $\mu$L of 5% (vol/vol) ammonium hydroxide into 20 $\mu$L of 20% (vol/vol) TFA for 15 min at 1,200 rpm. In the second elution, phosphopeptides were eluted with 70 $\mu$L of 20% (vol/vol) ammonium hydroxide in 60% (vol/vol) acetonitrile (pH 10.5). Finally, phosphopeptides were eluted with 20 $\mu$L of 60% ACN in 1% TFA. Acetonitrile was removed from the eluates by vacuum centrifugation, and

samples were acidified with TFA to pH 2, if necessary, and purified by StageTips. StageTips were prepared using an in-house protocol (68) as described previously (67).

**LC-MS/MS measurements.** Eluted peptides were separated by an Easy-nLC II system (Proxeon Biosystems) coupled online to an Orbitrap Elite mass spectrometer (Thermo Scientific) through a nanoelectrospray ion source (Proxeon Biosystems). Chromatographic separation was performed on a 2 cm fused silica analytical column with an inner diameter of 75 $\mu$m, packed in-house with reversed-phase ReproSil-Pur $C_{18}$-AQ 1.9 $\mu$m particles (Dr. Maisch GmbH). The column temperature was maintained at 40°C using an in-house-integrated column. Peptides were loaded onto the column with solvent A at a 700 nL/min flow rate using a maximum back pressure of 50,000 kPa. Peptides were eluted using a 60-min (for phosphoproteome analyses) or 120 min segmented gradient (for proteome analyses). The gradient was generated by 10 to 50% of solvent B at 40°C at a constant flow rate of 200 nL/min (67). The mass spectrometer was operated in a data-dependent mode, switching automatically between one full scan and subsequent MS/MS scans of either 12 (Top12 method) or 7 (Top7 method, phosphoproteome measurement) most abundant peaks selected with an isolation window of 1.4 *m/z* (mass/charge ratio). Full-scan MS spectra were acquired in a mass range from 300 to 1,650 *m/z* at a target value of $3 \times 10^6$ charges, with a maximum injection time of 25 ms and a resolution of 120,000 (defined at *m/z* 200). The higher-energy collisional dissociation MS/MS spectra were recorded with the maximum injection time of 45 or 220 ms (for phosphoproteome measurement) at a target value of $1 \times 10^5$ and a resolution of 30,000 (defined at *m/z* 200) or 60,000 for phosphoproteome measurement. The normalized collision energy was set to 27%, and the intensity threshold was kept at $1 \times 10^5$ or $5 \times 10^4$ for phosphoproteome measurement. The masses of sequenced precursor ions were dynamically excluded from MS/MS fragmentation for 30 s. Ions with unassigned charge states, with six or higher charged states, and singly charged were excluded from fragmentation selection. A few experiments were measured on an Orbitrap Elite mass spectrometer (Thermo Fisher Scientific) with previously described parameters (63).

**Raw data processing.** Acquired raw data were processed using the MaxQuant software suite version 2.0.3.0 (69). MaxQuant-generated peak lists were searched using the Andromeda search engine (70) against the proteome of *S. rimosus* G7 genome (GenBank accession number CP048261), as well as the associated plasmid (GenBank accession number CP048262). Carbamidomethylation of cysteines was set as a fixed modification, while methionine oxidation, protein N-terminal acetylation, and Ser/Thr/Tyr phosphorylation were defined as variable modifications. Trypsin was chosen as a protease with up to two missed cleavages, and the minimum required peptide length was set to seven amino acids. The mass tolerance window for parent and fragment ion isolation was set at 4.5 ppm and 20 ppm, respectively. A false-discovery rate (FDR) of 1% was set at the peptide, protein, and phosphorylation site levels (71).

**Statistical analyses.** Significance analyses of differentially regulated proteins and phosphorylation events were performed with Perseus software (version 1.6.14.0.). All contaminants, reverse hits, and proteins identified only by the modification site were removed. For quantitative proteome analyses, data were filtered to have at least one measured ratio in both replicates. Significantly regulated proteins were identified by one sample *t* test with a *P* value of 0.05 and an $s^0$ parameter of 0.1. For the phosphoproteome analyses, phosphorylation site ratios were normalized to their respective abundances of protein groups, transformed to $\log_2$, and plotted against the respective $\log_{10}$-transformed phosphorylation site intensities (72). An arbitrary threshold of 1.5 on a $\log_2$ scale (3-fold change) was used for the assignment of differentially regulated phosphorylation sites to certain developmental stages (if the phosphorylation site was quantified in all three stages).

**Clustering analysis.** Proteins with similar temporal abundances were clustered using hierarchical clustering analysis. This analysis was performed using Perseus on significantly regulated proteins obtained by *t* test (*P* < 0.05). Proteins were normalized using Z-score and clustered based on calculation of Euclidean distance.

**Functional annotation and functional enrichment analysis.** The proteins were functionally annotated with gene ontology (GO) biological processes (GOBP), molecular functions (GOMF), cellular compartments (GOCC), Kyoto Encyclopedia of Genes and Genomes (KEGG), protein families (PFAM), and enzyme commission (EC) functional resources and added to Perseus software. These annotations were obtained by performing protein sequence orthologous assignment with the evolutionary genealogy of genes: nonsupervised orthologous groups (eggNOG-mapper) software (73). Proteins with differentially regulated phosphorylation sites were grouped into six major functional categories according to their biological roles, as the number of phosphoproteins per 20 subcategories defined by eggNOG was relatively small. Clusters were tested for enrichment based on the mentioned annotation terms by using Fisher's exact test. Annotation terms with adjusted *P* values below 0.05 in any of the clusters and with an enrichment factor above 1 were considered significant.

**Phylogenetic analysis.** Predicted Ser/Thr kinases (STPKs) from the *S. rimosus* G7 strain (32 sequences) and 25 out of 27 STPKs of *S. coelicolor* M415 identified by other proteomics studies (15, 16) were used to perform multiple-sequence alignment (MSA) with the PROMALS3D tool (74). Two kinases were removed, SCO3360, which was annotated as a hypothetical protein which shared similarity to aminoglycoside 3′-phosphotransferase, and SCO3941, which was biochemically characterized as a protein phosphatase with dual-substrate specificity (75). Two STPKs with solved three-dimensional structures from *Mycobacterium tuberculosis* (PDB IDs 6I2P and 3OUV) were used to improve alignment quality. A phylogenetic tree of aligned kinase sequences was constructed using the maximum-likelihood method in PhyML (76). Approximate likelihood ratio test (aLRT) values were used to infer branch support. Branches with aLRT values greater than 0.9 were considered well supported.

**Analysis of phospho-orthologs in prokaryotes.** To identify *S. rimosus* phospho-orthologs in other prokaryotes, we used the dbPSP 2.0 database (http://dbpsp.biocuckoo.cn). It comprises 19,296

experimentally identified phosphorylation sites in 8,586 proteins from 200 microorganisms that belong to 12 phyla within domains of *Bacteria* and *Archaea*.

**Data availability.** The mass spectrometry proteomics data have been deposited to the ProteomeXchange Consortium via the PRIDE (77) partner repository with the data set identifier PXD034565. All other data supporting the findings of this study are available in the article or the supplemental material.

## SUPPLEMENTAL MATERIAL

Supplemental material is available online only.

**DATA SET S1**, XLSX file, 2.9 MB.
**FIG S1**, TIF file, 0.4 MB.
**FIG S2**, TIF file, 0.2 MB.
**FIG S3**, TIF file, 0.1 MB.
**TABLE S1**, DOCX file, 0.01 MB.
**TABLE S2**, DOCX file, 0.01 MB.
**TABLE S3**, DOCX file, 0.01 MB.
**TABLE S4**, DOCX file, 0.01 MB.
**TABLE S5**, DOCX file, 0.04 MB.

## ACKNOWLEDGMENTS

We thank Marin Roje and Mladenka Jurin for the measurement of OTC in culture media, Božidar Šantek and Marina Grubić for the measurement of glucose consumption, and Ivo Crnolatac for valuable discussion.

This work was supported by the Unity through Knowledge Fund (UKF grant 27/15 to D.V.). B.M. was supported by grants from the Deutsche Forschungsgemeinschaft (German Research Foundation Cluster of Excellence EXC 2124, SFB 766, FOR 2816, TRR 261; project ID 398967434), the European Union's Horizon 2020 Research and Innovation Program under the Marie Sklodowska-Curie grant agreement 955626, and the German-Israeli Foundation grant I-1464-416.13/2018. E.Š. was supported by the UKF fund and by the Croatian Government and the EU through the ERDF's Competitiveness and Cohesion Operational Program (KK.01.1.1.01), The Scientific Centre of Excellence for Marine Bioprospecting (BioProCro).

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
