## [Reviewer comments · mSystems]

Phosphoproteome dynamics of *Streptomyces rimosus* during submerged growth and antibiotic production

Ela Šarić, Gerry Quinn, Nicolas Nalpas, Tina Paradžik, Saša Kazazić, Želimira Filić, Maja Šemanjski, Paul Herron, Iain Hunter, Boris Macek, and Dušica Vujaklija

Corresponding Author(s): Dušica Vujaklija, Ruđer Bošković Institute

Review Timeline:

Submission Date:	March 1, 2022
Editorial Decision:	March 31, 2022
Revision Received:	May 19, 2022
Editorial Decision:	June 13, 2022
Revision Received:	June 21, 2022
Accepted:	July 13, 2022

Editor: Joshua Elias

Reviewer(s): Disclosure of reviewer identity is with reference to reviewer comments included in decision letter(s). The following individuals involved in review of your submission have agreed to reveal their identity: Marie-Joelle VIROLLE (Reviewer #1)

Transaction Report:

DOI: <https://doi.org/10.1128/msystems.00199-22>

March 31, 2022

Prof. Dusica Vujaklija
Rudjer Boskovic Institute
Molecular Biology
Bijenicka 54
Zagreb 10000
Croatia

Re: mSystems00199-22 (Phosphoproteome dynamics of *Streptomyces rimosus* during submerged growth and antibiotic production)

Dear Prof. Dusica Vujaklija:

Thank you for submitting your manuscript to mSystems. We have completed our review and I am pleased to inform you that, in principle, we expect to accept it for publication in mSystems. However, acceptance will not be final until you have adequately addressed the reviewer comments.

Preparing Revision Guidelines

Please include a Data Availability section as described here: <https://journals.asm.org/open-data-policy>

Please also reduce the number of supplementary display items. For complete guidelines on revision requirements, please see the journal Submission and Review Process requirements at <https://journals.asm.org/journal/mSystems/submission-review-process>. **Submission of a paper that does not conform to mSystems guidelines will delay acceptance of your manuscript.**

Sincerely,

Joshua Elias

Editor, mSystems

Journals Department
Reviewer comments:

Reviewer #1 (Comments for the Author):

In their paper entitled « Phosphoproteome dynamics of *Streptomyces rimosus* during submerged growth and antibiotic production" Saric et al. provide a list of phosphorylated proteins detected at three different time points of the growth of *S. rimosus* in liquid cultures. This strain is of industrial interest as a main oxytetracycline producer and even in the native strain under study, oxytetracycline production seems concomitant to active growth. In contrast to other *Streptomyces* species, *S. rimosus* grows in a rather dispersed form that might result from the occurrence of a cell death process. The identified phosphorylated proteins identified in the course of this study were classified in different ontological groups (Figure 2) and the variation of their abundance was assessed throughout growth (Figure 3). Five different abundance patterns were proposed (Figure 4) and their content was commented and compared to similar data obtained in *S. coelicolor*. This paper that is mainly descriptive is rather well written (although a bit long...) and could be of interest for scientists in the field. The main criticism that can be made to this study is the small number of replicates (2) that does not really allow meaningful statistical analysis. Furthermore, the sentence in line 432 is deeply wrong and should be modified as "To determine protein phosphorylation concomitant to the increase in OTC production and pellet fragmentation". Indeed this study does not provide any causal relationships between protein phosphorylation and the mentioned processes. In line 384 it would be better to write : "... this observation determined the choice of the first growth point..." At last in line 386, it seems that the term "transition phase" might not have the same meaning in this manuscript and in other papers. In most papers the transition phase is a phase of growth slow-down/arrest between two phases of active growth, the second one being slower than the first one (in the case of this manuscript it would be SI), whereas the authors call transition phase the entry into stationary phase (phase SII). It would be nice to clarify this point. Other general comments: It would be interesting to determine whether the production of oxytetracycline, that inhibits protein synthesis, impacts cellular growth by comparing growth of the native strain with that of a strain deleted for the oxytetracycline cluster? Would oxytetracycline production have an impact on the cellular death/lysis process ? Could it be the cause or the consequence of the cellular death/lysis process?

Reviewer #2 (Comments for the Author):

The manuscript by Saric et al., describes phosphoproteome of industrially important bacterium *Streptomyces rimosus*, producer of broad spectrum antibiotic oxytetracycline. Authors studied natural isolate of *S. rimosus*, which is parent strain of industrial *S. rimosus* strains used in fermentation industry. Authors identified over 417 phosphorylation sites from 230 phosphoproteins, out of (total) 3725 proteins which were identified in this study. Based on the genome annotation, this represents over 40% of the entire proteome. The entire phosphoproteome of *S. rimosus* was compared to the model streptomycete *Streptomyces coelicolor* phosphoproteome).

General comments

1. The technical approach and phosphoproteome analysis carried/used in this study is sound.
2. Authors correlated phosphoproteome analysis to mycelium morphology and antibiotic oxytetracycline production. This is an important observation, considering it relates to the selection of sampling points. It is very interesting how *S. rimosus* forms different morphological shapes of mycelium in the liquid culture. The formation of pellets is observed already very early during the process. However, it seems, that pellets are disintegrated already after 40 hours of incubation?

Question: Is the disintegration of *S. rimosus* mycelia pellets result of glucose consumption? Is glucose consumed completely soon after 40 hours of incubation? Or, is the disintegration of mycelia really result of morphological changes, as a property of *S. rimosus*?

I believe, it would be very useful to have an answer on these questions, considering that authors divided sampling points for

proteome analysis based on these observations. For this purpose, I believe, it would be very useful to present glucose consumption in the Figure 1A.

3. Authors also correlate sampling points in relation to antibiotic oxytetracycline production. Which is clearly indicated in the title of the manuscript. However, it is not easy, based on the data presented in the text, what is the dynamics of oxytetracycline production in relation to the morphological development/pellet formation and glucose consumption. Therefore, if possible, I would suggest to present antibiotic titer on the Figure 1A, in addition to glucose concentration.

4. As mentioned earlier, authors have completed very sound analysis of phosphoproteome accompanied with thorough analysis of the generated data. However, total proteome analysis did not really detect many proteins involved in biosynthesis of secondary metabolites. Is this expected? Could authors discuss this ?

Although biosynthesis of oxytetracycline was detected during the fermentation, not even enzymes involved in biosynthesis of oxytetracycline were identified. On the other hand, a number of regulatory proteins, likely produced at very low concentrations were successfully detected. Therefore, considering that so few proteins catalysing secondary metabolite biosynthesis were detected, it would be perhaps useful to discuss in the manuscript:

- a) Is perhaps cultivation in the liquid culture, in a glucose rich medium reason for poor secondary metabolite production? How could this impact results on analysis of phospho(proteome)?
- b) Or is perhaps proteome analysis of enzymes involved in secondary metabolite production somehow limited?

Considering the title of the manuscript, it would be useful to add additional discussion related to this question and in correlation to points 2, 3 and 4.

5. Considering that the samples were selected at three time points, would it be possible to present more information on phosphorylation status of phosphorylated proteins at the different time points? It would be useful, if possible, to correlate this phosphorylation status by the onset of antibiotic oxytetracycline production. These kind of data would likely be very useful starting point for future studies.

Minor comments:

Page 3, line 61: oxytracycline

Figure S2 (to explain better the figure- outer/inner cycle)

Figure S3 (to explain better the figure- outer/inner cycle)

Point-by-point responses to the questions raised by the reviewers:

Reviewer comments:

Reviewer #1 (Comments for the Author):

In their paper entitled « Phosphoproteome dynamics of *Streptomyces rimosus* during submerged growth and antibiotic production" Saric et al. provide a list of phosphorylated proteins detected at three different time points of the growth of *S. rimosus* in liquid cultures. This strain is of industrial interest as a main oxytetracycline producer and even in the native strain under study, oxytetracycline production seems concomitant to active growth. In contrast to other *Streptomyces* species, *S. rimosus* grows in a rather dispersed form that might result from the occurrence of a cell death process.

The identified phosphorylated proteins identified in the course of this study were classified in different ontological groups (Figure 2) and the variation of their abundance was assessed throughout growth (Figure 3). Five different abundance patterns were proposed (Figure 4) and their content was commented and compared to similar data obtained in *S. coelicolor*.

This paper that is mainly descriptive is rather well written (although a bit long...) and could be of interest for scientists in the field.

Reviewer #1 (R#1): The main criticism that can be made to this study is the small number of replicates (2) that does not really allow meaningful statistical analysis.

Answer (A): We agree with the referee that accumulation and analysis of more replicates would allow for a more elaborate statistical analysis. However, MS measurements of *S. rimosus* samples turned out to be very demanding due to interference of other secondary metabolites. We therefore opted to perform a direct quantitative comparison of different growth stages using stable isotope (dimethyl) labeling, which enables precise relative quantification and minimizes the number of missing values in the dataset. As a minimum, we performed two biological replicates and verified that they showed a good correlation. We note that our limited number of replicates does not negate our current findings regarding significantly changing proteins and phosphorylation sites (1). This approach was used by us and others in cases where multiple replicates were not feasible (2-4).

References

1. Goulet, M.-A. and Cousineau, D. (2019) The Power of Replicated Measures to Increase Statistical Power. *Advances in Methods and Practices in Psychological Science*, **2**, 199-213.
2. Spat, P., Klotz, A., Rexroth, S., Macek, B. and Forchhammer, K. (2018) Chlorosis as a Developmental Program in Cyanobacteria: The Proteomic Fundament for Survival and Awakening. *Mol. Cell. Proteomics*, **17**, 1650-1669.
3. Zittlau, K.I., Terradas, A.L., Nalpas, N., Geisler, S., Kahle, P.J. and Macek, B. Temporal Analysis of Protein Ubiquitylation and Phosphorylation During Parkin-dependent Mitophagy. *Mol. Cell. Proteomics*.

4. Mertins, P., Udeshi, N.D., Clauser, K.R., Mani, D.R., Patel, J., Ong, S.E., Jaffe, J.D. and Carr, S.A. (2012) iTRAQ labeling is superior to mTRAQ for quantitative global proteomics and phosphoproteomics. *Mol. Cell. Proteomics*, **11**, M111 014423.

R#1: Furthermore, the sentence in line 432 is deeply wrong and should be modified as "To determine protein phosphorylation concomitant to the increase in OTC production and pellet fragmentation".

Indeed, this study does not provide any causal relationships between protein phosphorylation and the mentioned processes.

A: We thank the reviewer for this observation. We completely agree with him and we have changed the sentence in the text as suggested line 432 (now line 413).

R#1: In line 384 it would be better to write : "... this observation determined the choice of the first growth point..."

A: We thank the reviewer for this suggestion; the sentence (line 378) has been changed accordingly (now line 381).

At last in line 386, it seems that the term "transition phase" might not have the same meaning in this manuscript and in other papers.

R#1: In most papers the transition phase is a phase of growth slow-down/arrest between two phases of active growth, the second one being slower than the first one (in the case of this manuscript it would be SI), whereas the authors call transition phase the entry into stationary phase (phase SII). It would be nice to clarify this point.

A: In order to clarify growth points (SI, SII and SIII), for the SI growth point in the text of the manuscript we have stated that it is „arrest phase“, for SII we stated „when growth slowed and bacteria began to enter the stationary phase“ (line 167), “at the beginning of stationary phase” (line 379) and „the entry into stationary phase“ (line 615), while point SIII remained „late stationary phase“.

R#1: Other general comments:

It would be interesting to determine whether the production of oxytetracycline, that inhibits protein synthesis, impacts cellular growth by comparing growth of the native strain with that of a strain deleted for the oxytetracycline cluster?

A: This is exactly the kind of experiment that was done for the bacterium *Streptomyces griseus* (1). *S. griseus* produces streptomycin, which also inhibits protein translation. Neuman T. *et al* (1) reported that wild type *S. griseus* and strain(s) deficient in Sm production show the same growth characteristics (i.e. the beginning of growth arrest phase and the beginning of the stationary phase overlap in time between two examined strains).

We believe that a similar result would be obtained for the proposed experiment especially considering that our experiments were conducted with a wild-type strain that produces low concentrations of OTC. Like other antibiotic-producing bacteria, *S. rimosus* contains determinants of resistance (in this case *otrA* and *otrB*) that map to the *otc* gene cluster and contribute to OTC self-resistance (e.g., OtrA protects ribosomes, while OtrB is responsible for OTC efflux). In earlier work McDowall et al (2) concluded, 'the *otrA* resistance gene being cotranscribed with *otcC* as part of a polycistronic message, suggesting a simple mechanism of coordinate regulation which ensures that resistance to the antibiotic increases in proportion to production'.

We therefore believe that *S. rimosus* will be phenotypically-resistant to OTC throughout the experiments described and hence OTC will not affect translation.

References:

1. Neuman T., Piepersberg, W., Diestler, J. (1996) Decision phase of streptomycin production in *Streptomyces griseus*. *Microbiology*, 142, 1953-1963
2. McDowall, K. J., Thamchaipenet, A., Hunter, I.S., Phosphate Control of Oxytetracycline Production by *Streptomyces rimosus* Is at the Level of Transcription from Promoters Overlapped by Tandem Repeats Similar to Those of the DNA-Binding Sites of the OmpR Family, *Journal of Bacteriology*, 181, No. 10

R#1:(i) Would oxytetracycline production have an impact on the cellular death/lysis process?
(ii) Could it be the cause or the consequence of the cellular death/lysis process?

A: (i) Taking into account the growth curve presented above and described self-defense mechanisms of antibiotic producer, *S. rimosus* as well as the fact that a wild-type strain used in our study does not produce large amounts of antibiotics, it is not to be expected that the concentrations of antibiotic achieved in this work will have effect on cellular death/lysis process.

(ii) Growth arrest is characterized by activation of antibiotic synthesis (1), reduction of ribosomal proteins (2), and stress response (3). However, it has been reported that the antibiotic is detected at the end of growth arrest (4), while cell death has been reported to begin before that growth phase and that new antibiotic-producing mycelium develops from the remaining hyphae (5).

In our work, during growth arrest (SI) the most abundant were proteins which correspond to the occurrence of cell death which then intensifies (cold-shock, GroES, CarD regulator DNA-helicase RecQ, DEAD/DEAH box helicase). Thus, it was not unexpected that this group of proteins was retained in the SII phase (please see Fig. 4 cluster C and Data Set S1 - Sheet 4). However, proteins related to secondary and lipid metabolism appeared to be more abundant during SII phase (see Fig. 4 cluster B). Moreover, with less stringent criteria (the arbitrary cut off value 1.5 on a log₂ scale; Data Set S1 - Sheet 3) proteins from the OTC cluster (13/24) were mostly upregulated during SII/SIII phase (please see Table below). Note that this information is provided in Supplementary (Date Set S1, Sheet 3.) but we have extracted it here to clarify our points).

Based on these results, we believe that the appearance of antibiotics is the consequence of the cellular death/lysis process: however, addressing this question is outside of the scope of this paper.

Proteins from OTC cluster in G7 (13/24):

Locus tag	Protein name	Significantly regulated in (with a 1.5 cut off on a log ₂ scale)
SRIM_038070	Methyltransferase, oxyT	rep 1 SIII, rep2 SII and SIII
SRIM_038075	hypothetical protein, oxyS	rep 1 SIII, rep2 SII and SIII
SRIM_038080	PPOX class F420-dependent oxidoreductase, oxyR	rep 1 SIII, rep2 SIII
SRIM_038100	cyclase family protein, oxyN	ratio not measured
SRIM_038105	SDR family oxidoreductase, oxyM	rep2 SIII
SRIM_038115	tetracycline nonaketamide D-ring cyclase/dehydratase OxyK	rep 1 SIII, rep2 SIII
SRIM_038120	SDR family NAD(P)-dependent oxidoreductase, oxyJ	ratio not measured
SRIM_038125	nuclear transport factor 2 family protein, oxyI	rep 1 SIII, rep2 SIII
SRIM_038130	AMP-binding protein, oxyG/oxyH	rep 1 SIII, rep2 SII and SIII
SRIM_038135	Methyltransferase, oxyF	rep 1 SIII, rep2 SII and SIII
SRIM_038150	acyl carrier protein, oxyC	ratio not measured
SRIM_038160	tetracycline polyketide synthase beta-ketoacyl synthase subunit OxyA	ratio not measured
SRIM_038170	oxytetracycline resistance efflux MFS transporter OtrB	not significantly increased

References:

- Holt, T.G., Chang, C., Laurent-Winter, C., Murakami, T., Garrels, J.I., Davies, J.E., Thompson, C.J., (1992) Global changes in gene expression related to antibiotic synthesis in *Streptomyces hygroscopicus*. *Mol Microbiol.*, 6(8):969-80
- Blanco G, Rodicio MR, Puglia AM, Méndez C, Thompson CJ, Salas JA. (1994) Synthesis of ribosomal proteins during growth of *Streptomyces coelicolor*. 12(3):375-85.
- Puglia, A.M., Vohradsky, J., Thompson C.J., (1995) Developmental control of the heat-shock stress regulon in *Streptomyces coelicolor*. *Mol Microbiol.*,17(4):737-46
- Neuman T., Piepersberg, W., Diestler, J. (1996) Decision phase of streptomycin production in *Streptomyces griseus*. *Microbiology*, 142, 1953-1963
- Manteca A, Alvarez R, Salazar N, Yagüe P, Sanchez J. (2008) Mycelium Differentiation and Antibiotic Production in Submerged Cultures of *Streptomyces coelicolor*. 74(12):3877-86

Reviewer #2 (Comments for the Author):

The manuscript by Saric et al., describes phosphoproteome of industrially important bacterium *Streptomyces rimosus*, producer of broad spectrum antibiotic oxytetracycline. Authors studied natural isolate of *S. rimosus*, which is parent strain of industrial *S. rimosus* strains used in fermentation industry. Authors identified over 417 phosphorylation sites from 230 phosphoproteins, out of (total) 3725 proteins which were identified in this study. Based on the genome annotation, this represents over 40% of the entire proteome. The entire phosphoproteome of *S. rimosus* was compared to the model streptomycete *Streptomyces coelicolor* phosphor(proteome).

General comments

1. The technical approach and phospho(proteome) analysis carried/used in this study is sound.
2. Authors correlated phosphoproteome analysis to mycelium morphology and antibiotic oxytetracycline production. This is important observation, considering it relates to the selection of sampling points.

It is very interesting how *S. rimosus* forms different morphological shapes of mycelium in the liquid culture. The formation of pellets is observed already very early during the process. However, it seems, that pellets are disintegrated already after 40 hours of incubation?

Question (Q): (i) Is the disintegration of *S. rimosus* mycelia pellets result of glucose consumption? Is glucose consumed completely soon after 40 hours of incubation? Or, is the disintegration of mycelia really result of morphological changes, as a property of *S. rimosus*?

(ii) I believe, it would be very useful to have answer on these questions, considering that authors divided sampling points for proteome analysis based on these observations. For this purpose, I believe, it would be very useful to present glucose consumption in the Figure 1A.

(i) Answer (A): Please see revised Figure 1A (below). Our data show that mycelial degradation observed after 40 hours of growth in *S. rimosus* is not the result of glucose depletion. We assume that change in mycelial morphology is a result of the complex developmental programme in *S. rimosus*. During the transition lag phase (SI), depletion of some preferential nutrients leads to growth arrest prior to antibiotic production (1). This phase is characterized by activation of proteins involved in the stress response (2) and reduction of ribosomal proteins (3), as well as activation of antibiotic resistance (4) and antibiotic biosynthesis genes (5)

(We discuss this phase of growth in the text in more details, please see lines 363-374)

Mycelial differentiation and production of antibiotic in liquid culture was published for the model organism *S. coelicolor*, but these two bacteria exhibit significant differences in pellet morphology and size. *S. coelicolor*, produces much larger pellets during growth in liquid medium, ~600 μm (6). In contrast, confocal microscopy analysis of bacterial cultures from multiple growth phases showed that *S. rimosus* produces much smaller pellets (~200 μm). The process of pellet disintegration and fragment formation during the later growth phase was observed during this study in wild-type G7 and recently in a strain producing 35X more OTC and it is a genetic property of *S. rimosus*. Little is known about this process, but a recent publication has shown that fragmentation in submerged cultures of *S. lividans* is initiated when cultures enter a stationary phase (7, 8); we believe that the same process occurs in *S. rimosus*.

(ii) A: According to the reviewer's suggestion, we analyzed the glucose consumption during the growth of *S. rimosus* and changed Fig. 1A accordingly (please see below). Note that there is ~ 35 g/L of glucose remaining at the end of the fermentation. This reviewer may be curious as to why the initial glucose concentration (50 g /L) was therefore so high. This is because this medium was also to be used for analysis of higher OTC producers, where

considerably more of the (glucose) carbon flux ends up in OTC. This is a subject for the follow up paper, the project is still ongoing but the present paper was to establish the baseline for the soil isolate / low producer. (please lines 404 - 408)

References:

- Holt, T.G., Chang, C., Laurent-Winter, C., Murakami, T., Garrels, J.I., Davies, J.E., Thompson, C.J., (1992) Global changes in gene expression related to antibiotic synthesis in *Streptomyces hygroscopicus*. *Mol Microbiol.*, 6(8):969-80
- Puglia, A.M., Vohradsky, J., Thompson C.J., (1995) Developmental control of the heat-shock stress regulon in *Streptomyces coelicolor*. *Mol Microbiol*,17(4):737-46
- Blanco G, Rodicio MR, Puglia AM, Méndez C, Thompson CJ, Salas JA. (1994) Synthesis of ribosomal proteins during growth of *Streptomyces coelicolor*. 12(3):375-85.
- Salah-Bey K., Blanc, V., Thompson C.J., (1995) Stress-activated expression of a *Streptomyces pristinaespiralis* multidrug resistance gene (*ptr*) in various *Streptomyces* spp. and *Escherichia coli*. *Mol. Microbiol.* 17:1001–1012.
- Huang, J., C. J. Lih, K. H. Pan, and S. N. Cohen. (2001) Global analysis of growth phase responsive gene expression and regulation of antibiotic biosynthetic pathways in *Streptomyces coelicolor* using DNA microarrays. *Genes Dev.* 15:3183–3192.
- Manteca A, Alvarez R, Salazar N, Yagüe P, Sanchez J. (2008) Mycelium Differentiation and Antibiotic Production in Submerged Cultures of *Streptomyces coelicolor*. 74(12):3877-86
- vanDiesel, D., Claessen, D., vanWezel G.P., (2014) Chapter One - Morphogenesis of *Streptomyces* in Submerged Cultures, *Advances in Appl. Microbiol.*, 89, 1-45
- Zacchetti B., Smits P., Claessen D., (2018) Dynamics of Pellet Fragmentation and Aggregation in Liquid-Grown Cultures of *Streptomyces lividans*. *Front. Microbiol.* 9:943

3. Authors also correlate sampling points in relation to antibiotic oxytetracycline production. Which is clearly indicated in the title of the manuscript. However, it is not easy, based on the data presented in the text, what is the dynamics of oxytetracycline production in relation to the morphological development/pellet formation and glucose consumption. Therefore, if possible, I would suggest to present antibiotic titer on the Figure 1A, in addition to glucose concentration.

A: According to the reviewer's suggestion in addition to the glucose consumption we also present antibiotic titer in revised Figure 1A (please see below).

Fig. 1A

4. (R#2) As mentioned earlier, authors have completed very sound analysis of phosphoproteome accompanied with thorough analysis of the generated data. However, total proteome analysis did not really detect many proteins involved in biosynthesis of secondary metabolites. Is this expected? Could authors discuss this?

A: With thank the reviewer for this comment and in order to clarify it all proteins assigned by eggNOG mapper to functional category “Secondary metabolites biosynthesis, transport and catabolism” were highlighted in our proteome table for readers interested in this functional category (please see Data Set1, Sheet 1, proteins marked in grey). Note that ~ 300 proteins in the whole proteome of *S. rimosus* were assigned to this functional category and out of them we have detected 140 proteins. Nevertheless, we agree with the reviewer, only ~50% of proteins involved in biosynthesis of secondary metabolite (OTC) were detected during this study. However, despite the high concentration of glucose, the bacteria exhibited typical multiphase growth as well as the appearance of OTC at expected point of growth, during the growth arrest phase (for details please see explanation above – for Q2). OTC biosynthesis was at low levels and only 13/24 proteins involved in the biosynthetic pathway were detected in the proteome but not at statistically significant levels. Most of them (8/24) were detected during the SII and SIII phases when there is an increase in OTC in the medium. We are sorry if we did not clarify this but this information is now highlighted in the manuscript (see Table below; Note that these proteins are now marked in Supplementary, Data Set 1, Sheet3 as well). As stated above, for comparison, the high (50 g/L) concentration of glucose in this medium was standardized for low- and high-producers. It is probable that the higher concentration of glucose here repressed the expression of OTC production genes, which is likely why the levels of these gene products were so low in this proteomic analysis. Correlating with this, in the higher production strain (23383) when more glucose was utilized, 22/24 of the OTC pathway proteins were detected (ongoing study) – to be discussed in a subsequent manuscript but below the list of these proteins is included in this response for information to this Referee.

Proteins identified in the qualitative analysis (OtcR, OtrB, OxyA, OxyB, OxyC, OxyD, OxyE, OxyF, OxyG/OxyH, OxyI, OxyJ, OxyK, OxyL, OxyM, OxyN, OxyP, OxyQ, OxyR, OxyS, OxyT, OtrA and OtcG) in the high OTC producer.

Proteins from OTC cluster in G7 (13/24):

Locus tag	Protein name	Significantly regulated in (with a 1.5 cut off on a log ₂ scale)
SRIM_038070	Methyltransferase, oxyT	rep 1 SIII, rep2 SII and SIII
SRIM_038075	hypothetical protein, oxyS	rep 1 SIII, rep2 SII and SIII
SRIM_038080	PPOX class F420-dependent oxidoreductase, oxyR	rep 1 SIII, rep2 SIII
SRIM_038100	cyclase family protein, oxyN	ratio not measured
SRIM_038105	SDR family oxidoreductase, oxyM	rep2 SIII
SRIM_038115	tetracycline nonaketamide D-ring cyclase/dehydratase OxyK	rep 1 SIII, rep2 SIII
SRIM_038120	SDR family NAD(P)-dependent oxidoreductase, oxyJ	ratio not measured
SRIM_038125	nuclear transport factor 2 family protein, oxyI	rep 1 SIII, rep2 SIII
SRIM_038130	AMP-binding protein, oxyG/oxyH	rep 1 SIII, rep2 SII and SIII
SRIM_038135	Methyltransferase, oxyF	rep 1 SIII, rep2 SII and SIII
SRIM_038150	acyl carrier protein, oxyC	ratio not measured
SRIM_038160	tetracycline polyketide synthase beta-ketoacyl synthase subunit OxyA	ratio not measured
SRIM_038170	oxytetracycline resistance efflux MFS transporter OtrB	not significantly increased

R#2 Although biosynthesis of oxytetracycline was detected during the fermentation, not even enzymes involved in biosynthesis of oxytetracycline were identified. On the other hand, a number of regulatory proteins, likely produced at very low concentrations were successfully detected. Therefore, considering that so few proteins catalyzing secondary metabolite biosynthesis were detected, it would be perhaps useful to discuss in the manuscript:

a) Is perhaps cultivation in the liquid culture, in a glucose reach medium reason for poor secondary metabolite production? How could this impact results on analysis of phospho(proteome)?

A: We hope that we have already answered the question about glucose and the media we are using. We also thank the reviewer for questions regarding proteins from the OTC cluster. It is obvious that we did not highlight them enough so they are also listed in the table we have shown for the answer to question no. 4. With respect to that we made some changes in the manuscript (please see lines 557-565) and we put information in which supplementary tables are the data of MS analyses related to proteins from OTC cluster. We have also highlighted them further in the supplementary tables themselves (please see lines 559 and 565).

The reviewer noted well that we discovered a lot of regulatory proteins (note that is was one of our major goals!). We assume that this can be attributed to the growth conditions and selected specific stages in which we collected the samples. As described above, the bacterium undergoes dynamic cellular processes during selected phases, in which cell lysis, pellet disintegration, and antibiotic synthesis occur continuously. In addition, some regulatory proteins control multiple cellular processes. For example, the predicted kinase, SRIM_017990 was found to be phosphorylated at all stages of growth during which time the bacteria continuously synthesized OTC. However, this kinase is closely related to AfsK which is known to regulate polar growth, hyphae branching, and chromosome replication initiation in addition to antibiotic control (please lines 731-732).

b) Or is perhaps proteome analysis of enzymes involved in secondary metabolite production somehow limited?

A: Each bacterial sample can produce some products that limit the analysis. The authors applied modern technologies for MS analysis of bacterial (phospho) proteomes, but during this study *S. rimosus* samples proved to be very demanding due to the interference of other secondary metabolites (e.g. pigments, extracellular matrix ...). The approach applied to solve this problem is stated below (Q5). Note that during the final growth phase when the bacterium also produced the highest amount of antibiotic and when the OTC cluster genes should be most active, the bacterium also intensively produced an extracellular matrix that possibly interfered with protein extraction and detection levels.

It is certain that the technology and that the level of detection will improve over time, but we hope that our results (230 phosphoproteins of which 80% have not yet been reported in streptomycetes) will hopefully contributed to an increase in the bacterial database of phosphoproteomes.

R#2 Considering the title of the manuscript, it would be useful to add additional discussion related to this question and in correlation to points 2, 3 and 4.

A: As we have stated in the answers to the previous questions, in some parts of the manuscript we have added clarifications, additional references and additional information about the proteins involved in OTC biosynthesis (please see above).

5. Considering that the samples were selected at three time points, would it be possible to present more information on phosphorylation status of phosphorylated proteins at the different time points? It would be useful, if possible, to correlate this phosphorylation status by the onset of antibiotic oxytetracycline production. These kind of data would likely be very useful starting point for future studies.

A: We agree with the reviewer that measuring more points would most likely increase the number of detected (phospho) proteins. However, due to interference of secondary metabolites (pigment, cell-matrix proteins, extracellular polymers) MS measurements of *S. rimosus* samples turned out to be very demanding. As stated to Reviewer #1, a direct quantitative comparison of different growth phases was made using stable isotope (dimethyl) labeling. This allowed us to accurately quantify and at the same time minimize the number of missing values in the data set. In order to gain the best possible insight into the dynamics of (phospho)proteome, we selected three distinct growth phases in which significant regulatory changes occur and bacteria exhibit the most pronounced morphological changes (antibiotic activation starts in (SI; for detail description of SI phase please above), bacteria enter into the stationary phase (SII) and late stationary phase (SIII)).

Note that Manteca and co-workers also performed MS analysis in three distinct phases of growth with the model organism (*S. coelicolor* – (phospho)proteomes) (1, 2).

In addition, it should be considered that the culture of *S. rimosus* is extremely heterogeneous. Especially during the SI/SII stages of growth, we have a population of older pellets that disintegrate and at the same time the young hyphae that still form new pellets. Consequently, a large number of proteins were found in both stages of growth. We believe the most interesting are proteins that were uniquely found in any of the selected stages and we focused on describing them.

Reference:

1. Manteca, A., Ye, J., Sánchez, J., Jensen, O.N., (2011) Phosphoproteome analysis of *Streptomyces* development reveals extensive protein phosphorylation accompanying bacterial differentiation. *J Proteome Res.*, 10(12):5481-92
2. Rioseras, B., Shliaha, P.V., Gorshkov, V., Yagüe, P., López-García, M.T., Gonzalez-Quiñonez, N., Kovalchuk, S., Rogowska-Wrzesinska, A., Jensen, O.N., Manteca, A., (2018) Quantitative Proteome and Phosphoproteome Analyses of *Streptomyces coelicolor* Reveal Proteins and Phosphoproteins Modulating Differentiation and Secondary Metabolism. *Mol Cell Proteomics*, 17(8):1591-1611

Minor comments:

R#2 Page 3, line 61: oxytracycline

A: We thank the reviewer for noticing this typo, it has been changed to „oxytetracycline“ (line 52).

R#2 Figure S2 (to explain better the figure- outer/inner cycle)

R#2: Figure S3 (to explain better the figure- outer/inner cycle)

A: We thank the reviewers for noticing the lack of images description. Note that these figures are now **Fig. S1A and B** and corresponding text is added to Fig. legends:

Figure S1. The distribution of identified (phospho)proteins into major functional categories in *S. rimosus* and *S. coelicolor*. The outer circles display the main functional categories obtained by eggNOG-mapper for *S. rimosus* (**A**) and *S. coelicolor* (**B**) proteomes while inner circle those identified for phosphoproteomes. Only (phospho)proteins with one assigned functional category are shown (i.e. approx. 93% of the proteome and 90% of the phosphoproteome for *S. rimosus* and approx. 95% of (phospho)proteome for *S. coelicolor*).

A request to change:

- **Figure S5 / now Figure S3:** In addition to answering reviewers' questions and the changes we have made to the manuscript, the authors would like to change the colours in the

Figure S5 (please see below) to facilitate the comparison of distributions of functional categories and their subcategories in different clusters.

Namely, in the revised figure different shades of one color denote subcategories that belong to one specific category. In this way, we have simplified the graphical presentation of the results and we hope that the reviewers and editor will agree with this change.

Figure S3. Distribution of functional subcategories in each cluster. Each bar represents the distribution of functional subcategories in each cluster of proteins with similar temporal abundances.

- In addition, since the number of figures in supplements is limited to 10, the authors request permission to remove **Fig. S1** (the Western blot) because it is a control experiment to which reviewers had no objection, as well as **Fig. S6** which is an interesting observation but not crucial for results in this manuscript.
- Please note that, we added some additional results to Fig. 1A. Consequently, we had to expand some specific parts in the text with a corresponding description of the results and add appropriate discussion. For this reason we have shortened other parts of the manuscript to the maximum, as can be seen in the marked version. All changes requested by reviewers are marked in red while other changes in the text are marked by Track Changes.

June 13, 2022

Prof. Dušica Vujaklija
Ruđer Bošković Institute
Molecular Biology
Bijenicka 54
Zagreb 10000
Croatia

Re: mSystems00199-22R1 (Phosphoproteome dynamics of *Streptomyces rimosus* during submerged growth and antibiotic production)

Dear Prof. Dušica Vujaklija:

Thank you for submitting your manuscript to mSystems. We have completed our review and I am pleased to inform you that, in principle, we expect to accept it for publication in mSystems. However, acceptance will not be final until you have adequately addressed one remaining editorial request from the initial submission relating to complying with ASM's data policy.

Below you will find instructions from the mSystems editorial office and comments generated during the review.

Preparing Revision Guidelines

Once again, I request that the authors include a Data Availability section as described here: <https://journals.asm.org/open-data-policy>. I recommend submitting raw mass spectrometry data to the PRIDE repository (proteomexchange.org) or a similar one.

Sincerely,

Joshua Elias

Editor, mSystems

Journals Department
Reviewer comments:

Point-by-point responses to the questions raised by the reviewers:

Reviewer comments:

Reviewer #1 (Comments for the Author):

In their paper entitled « Phosphoproteome dynamics of *Streptomyces rimosus* during submerged growth and antibiotic production" Saric et al. provide a list of phosphorylated proteins detected at three different time points of the growth of *S. rimosus* in liquid cultures. This strain is of industrial interest as a main oxytetracycline producer and even in the native strain under study, oxytetracycline production seems concomitant to active growth. In contrast to other *Streptomyces* species, *S. rimosus* grows in a rather dispersed form that might result from the occurrence of a cell death process.

The identified phosphorylated proteins identified in the course of this study were classified in different ontological groups (Figure 2) and the variation of their abundance was assessed throughout growth (Figure 3). Five different abundance patterns were proposed (Figure 4) and their content was commented and compared to similar data obtained in *S. coelicolor*.

This paper that is mainly descriptive is rather well written (although a bit long...) and could be of interest for scientists in the field.

Reviewer #1 (R#1): The main criticism that can be made to this study is the small number of replicates (2) that does not really allow meaningful statistical analysis.

Answer (A): We agree with the referee that accumulation and analysis of more replicates would allow for a more elaborate statistical analysis. However, MS measurements of *S. rimosus* samples turned out to be very demanding due to interference of other secondary metabolites. We therefore opted to perform a direct quantitative comparison of different growth stages using stable isotope (dimethyl) labeling, which enables precise relative quantification and minimizes the number of missing values in the dataset. As a minimum, we performed two biological replicates and verified that they showed a good correlation. We note that our limited number of replicates does not negate our current findings regarding significantly changing proteins and phosphorylation sites (1). This approach was used by us and others in cases where multiple replicates were not feasible (2-4).

References

1. Goulet, M.-A. and Cousineau, D. (2019) The Power of Replicated Measures to Increase Statistical Power. *Advances in Methods and Practices in Psychological Science*, **2**, 199-213.
2. Spat, P., Klotz, A., Rexroth, S., Macek, B. and Forchhammer, K. (2018) Chlorosis as a Developmental Program in Cyanobacteria: The Proteomic Fundament for Survival and Awakening. *Mol. Cell. Proteomics*, **17**, 1650-1669.
3. Zittlau, K.I., Terradas, A.L., Nalpas, N., Geisler, S., Kahle, P.J. and Macek, B. Temporal Analysis of Protein Ubiquitylation and Phosphorylation During Parkin-dependent Mitophagy. *Mol. Cell. Proteomics*.

4. Mertins, P., Udeshi, N.D., Clauser, K.R., Mani, D.R., Patel, J., Ong, S.E., Jaffe, J.D. and Carr, S.A. (2012) iTRAQ labeling is superior to mTRAQ for quantitative global proteomics and phosphoproteomics. *Mol. Cell. Proteomics*, **11**, M111 014423.

R#1: Furthermore, the sentence in line 432 is deeply wrong and should be modified as "To determine protein phosphorylation concomitant to the increase in OTC production and pellet fragmentation".

Indeed, this study does not provide any causal relationships between protein phosphorylation and the mentioned processes.

A: We thank the reviewer for this observation. We completely agree with him and we have changed the sentence in the text as suggested line 432 (now line 413).

R#1: In line 384 it would be better to write : "... this observation determined the choice of the first growth point..."

A: We thank the reviewer for this suggestion; the sentence (line 378) has been changed accordingly (now line 381).

At last in line 386, it seems that the term "transition phase" might not have the same meaning in this manuscript and in other papers.

R#1: In most papers the transition phase is a phase of growth slow-down/arrest between two phases of active growth, the second one being slower than the first one (in the case of this manuscript it would be SI), whereas the authors call transition phase the entry into stationary phase (phase SII). It would be nice to clarify this point.

A: In order to clarify growth points (SI, SII and SIII), for the SI growth point in the text of the manuscript we have stated that it is „arrest phase“, for SII we stated „when growth slowed and bacteria began to enter the stationary phase“ (line 167), “at the beginning of stationary phase” (line 379) and „the entry into stationary phase“ (line 615), while point SIII remained „late stationary phase“.

R#1: Other general comments:

It would be interesting to determine whether the production of oxytetracycline, that inhibits protein synthesis, impacts cellular growth by comparing growth of the native strain with that of a strain deleted for the oxytetracycline cluster?

A: This is exactly the kind of experiment that was done for the bacterium *Streptomyces griseus* (1). *S. griseus* produces streptomycin, which also inhibits protein translation. Neuman T. *et al* (1) reported that wild type *S. griseus* and strain(s) deficient in Sm production show the same growth characteristics (i.e. the beginning of growth arrest phase and the beginning of the stationary phase overlap in time between two examined strains).

We believe that a similar result would be obtained for the proposed experiment especially considering that our experiments were conducted with a wild-type strain that produces low concentrations of OTC. Like other antibiotic-producing bacteria, *S. rimosus* contains determinants of resistance (in this case *otrA* and *otrB*) that map to the *otc* gene cluster and contribute to OTC self-resistance (e.g., OtrA protects ribosomes, while OtrB is responsible for OTC efflux). In earlier work McDowall et al (2) concluded, 'the *otrA* resistance gene being cotranscribed with *otcC* as part of a polycistronic message, suggesting a simple mechanism of coordinate regulation which ensures that resistance to the antibiotic increases in proportion to production'.

We therefore believe that *S. rimosus* will be phenotypically-resistant to OTC throughout the experiments described and hence OTC will not affect translation.

References:

1. Neuman T., Piepersberg, W., Diestler, J. (1996) Decision phase of streptomycin production in *Streptomyces griseus*. *Microbiology*, 142, 1953-1963
2. McDowall, K. J., Thamchaipenet, A., Hunter, I.S., Phosphate Control of Oxytetracycline Production by *Streptomyces rimosus* Is at the Level of Transcription from Promoters Overlapped by Tandem Repeats Similar to Those of the DNA-Binding Sites of the OmpR Family, *Journal of Bacteriology*, 181, No. 10

R#1:(i) Would oxytetracycline production have an impact on the cellular death/lysis process?
(ii) Could it be the cause or the consequence of the cellular death/lysis process?

A: (i) Taking into account the growth curve presented above and described self-defense mechanisms of antibiotic producer, *S. rimosus* as well as the fact that a wild-type strain used in our study does not produce large amounts of antibiotics, it is not to be expected that the concentrations of antibiotic achieved in this work will have effect on cellular death/lysis process.

(ii) Growth arrest is characterized by activation of antibiotic synthesis (1), reduction of ribosomal proteins (2), and stress response (3). However, it has been reported that the antibiotic is detected at the end of growth arrest (4), while cell death has been reported to begin before that growth phase and that new antibiotic-producing mycelium develops from the remaining hyphae (5).

In our work, during growth arrest (SI) the most abundant were proteins which correspond to the occurrence of cell death which then intensifies (cold-shock, GroES, CarD regulator DNA-helicase RecQ, DEAD/DEAH box helicase). Thus, it was not unexpected that this group of proteins was retained in the SII phase (please see Fig. 4 cluster C and Data Set S1 - Sheet 4). However, proteins related to secondary and lipid metabolism appeared to be more abundant during SII phase (see Fig. 4 cluster B). Moreover, with less stringent criteria (the arbitrary cut off value 1.5 on a log₂ scale; Data Set S1 - Sheet 3) proteins from the OTC cluster (13/25) were mostly upregulated during SII/SIII phase (please see Table below). Note that this information is provided in Supplementary (Date Set S1, Sheet 3.) but we have extracted it here to clarify our points).

Based on these results, we believe that the appearance of antibiotics is the consequence of the cellular death/lysis process: however, addressing this question is outside of the scope of this paper.

Proteins from OTC cluster in G7 (13/24):

Locus tag	Protein name	Significantly regulated in (with a 1.5 cut off on a log ₂ scale)
SRIM_038070	Methyltransferase, oxyT	rep 1 SIII, rep2 SII and SIII
SRIM_038075	hypothetical protein, oxyS	rep 1 SIII, rep2 SII and SIII
SRIM_038080	PPOX class F420-dependent oxidoreductase, oxyR	rep 1 SIII, rep2 SIII
SRIM_038100	cyclase family protein, oxyN	ratio not measured
SRIM_038105	SDR family oxidoreductase, oxyM	rep2 SIII
SRIM_038115	tetracycline nonaketamide D-ring cyclase/dehydratase OxyK	rep 1 SIII, rep2 SIII
SRIM_038120	SDR family NAD(P)-dependent oxidoreductase, oxyJ	ratio not measured
SRIM_038125	nuclear transport factor 2 family protein, oxyI	rep 1 SIII, rep2 SIII
SRIM_038130	AMP-binding protein, oxyG/oxyH	rep 1 SIII, rep2 SII and SIII
SRIM_038135	Methyltransferase, oxyF	rep 1 SIII, rep2 SII and SIII
SRIM_038150	acyl carrier protein, oxyC	ratio not measured
SRIM_038160	tetracycline polyketide synthase beta-ketoacyl synthase subunit OxyA	ratio not measured
SRIM_038170	oxytetracycline resistance efflux MFS transporter OtrB	not significantly increased

References:

- Holt, T.G., Chang, C., Laurent-Winter, C., Murakami, T., Garrels, J.I., Davies, J.E., Thompson, C.J., (1992) Global changes in gene expression related to antibiotic synthesis in *Streptomyces hygroscopicus*. *Mol Microbiol.*, 6(8):969-80
- Blanco G, Rodicio MR, Puglia AM, Méndez C, Thompson CJ, Salas JA. (1994) Synthesis of ribosomal proteins during growth of *Streptomyces coelicolor*. *12(3):375-85*.
- Puglia, A.M., Vohradsky, J., Thompson C.J., (1995) Developmental control of the heat-shock stress regulon in *Streptomyces coelicolor*. *Mol Microbiol.*,17(4):737-46
- Neuman T., Piepersberg, W., Diestler, J. (1996) Decision phase of streptomycin production in *Streptomyces griseus*. *Microbiology*, 142, 1953-1963
- Manteca A, Alvarez R, Salazar N, Yagüe P, Sanchez J. (2008) Mycelium Differentiation and Antibiotic Production in Submerged Cultures of *Streptomyces coelicolor*. *74(12):3877-86*

Reviewer #2 (Comments for the Author):

The manuscript by Saric et al., describes phosphoproteome of industrially important bacterium *Streptomyces rimosus*, producer of broad spectrum antibiotic oxytetracycline. Authors studied natural isolate of *S. rimosus*, which is parent strain of industrial *S. rimosus* strains used in fermentation industry. Authors identified over 417 phosphorylation sites from 230 phosphoproteins, out of (total) 3725 proteins which were identified in this study. Based on the genome annotation, this represents over 40% of the entire proteome. The entire phosphoproteome of *S. rimosus* was compared to the model streptomycete *Streptomyces coelicolor* phosphoproteome).

General comments

1. The technical approach and phosphoproteome analysis carried/used in this study is sound.

2. Authors correlated phosphoproteome analysis to mycelium morphology and antibiotic oxytetracycline production. This is important observation, considering it relates to the selection of sampling points.

It is very interesting how *S. rimosus* forms different morphological shapes of mycelium in the liquid culture. The formation of pellets is observed already very early during the process. However, it seems, that pellets are disintegrated already after 40 hours of incubation?

Question (Q): (i) Is the disintegration of *S. rimosus* mycelia pellets result of glucose consumption? Is glucose consumed completely soon after 40 hours of incubation? Or, is the disintegration of mycelia really result of morphological changes, as a property of *S. rimosus*?

(ii) I believe, it would be very useful to have answer on these questions, considering that authors divided sampling points for proteome analysis based on these observations. For this purpose, I believe, it would be very useful to present glucose consumption in the Figure 1A.

(i) Answer (A): Please see revised Figure 1A (below). Our data show that mycelial degradation observed after 40 hours of growth in *S. rimosus* is not the result of glucose depletion. We assume that change in mycelial morphology is a result of the complex developmental programme in *S. rimosus*. During the transition lag phase (SI), depletion of some preferential nutrients leads to growth arrest prior to antibiotic production (1). This phase is characterized by activation of proteins involved in the stress response (2) and reduction of ribosomal proteins (3), as well as activation of antibiotic resistance (4) and antibiotic biosynthesis genes (5)

(We discuss this phase of growth in the text in more details, please see lines 363-374)

Mycelial differentiation and production of antibiotic in liquid culture was published for the model organism *S. coelicolor*, but these two bacteria exhibit significant differences in pellet morphology and size. *S. coelicolor*, produces much larger pellets during growth in liquid medium, ~600 μm (6). In contrast, confocal microscopy analysis of bacterial cultures from multiple growth phases showed that *S. rimosus* produces much smaller pellets (~200 μm). The process of pellet disintegration and fragment formation during the later growth phase

was observed during this study in wild-type G7 and recently in a strain producing 35X more OTC and it is a genetic property of *S. rimosus*. Little is known about this process, but a recent publication has shown that fragmentation in submerged cultures of *S. lividans* is initiated when cultures enter a stationary phase (7, 8); we believe that the same process occurs in *S. rimosus*.

(ii) A: According to the reviewer's suggestion, we analyzed the glucose consumption during the growth of *S. rimosus* and changed Fig. 1A accordingly (please see below). Note that there is ~ 35 g/L of glucose remaining at the end of the fermentation. This reviewer may be curious as to why the initial glucose concentration (50 g /L) was therefore so high. This is because this medium was also to be used for analysis of higher OTC producers, where considerably more of the (glucose) carbon flux ends up in OTC. This is a subject for the follow up paper, the project is still ongoing but the present paper was to establish the baseline for the soil isolate / low producer. (please lines 404 - 408)

References:

1. Holt, T.G., Chang, C., Laurent-Winter, C., Murakami, T., Garrels, J.I., Davies, J.E., Thompson, C.J., (1992) Global changes in gene expression related to antibiotic synthesis in *Streptomyces hygroscopicus*. *Mol Microbiol.*, 6(8):969-80
2. Puglia, A.M., Vohradsky, J., Thompson C.J., (1995) Developmental control of the heat-shock stress regulon in *Streptomyces coelicolor*. *Mol Microbiol*,17(4):737-46
3. Blanco G, Rodicio MR, Puglia AM, Méndez C, Thompson CJ, Salas JA. (1994) Synthesis of ribosomal proteins during growth of *Streptomyces coelicolor*. 12(3):375-85.
4. Salah-Bey K., Blanc, V., Thompson C.J., (1995) Stress-activated expression of a *Streptomyces pristinaespiralis* multidrug resistance gene (ptr) in various *Streptomyces* spp. and *Escherichia coli*. *Mol. Microbiol.* 17:1001–1012.
5. Huang, J., C. J. Lih, K. H. Pan, and S. N. Cohen. (2001) Global analysis of growth phase responsive gene expression and regulation of antibiotic biosynthetic pathways in *Streptomyces coelicolor* using DNA microarrays. *Genes Dev.* 15:3183–3192.
6. Manteca A, Alvarez R, Salazar N, Yagüe P, Sanchez J. (2008) Mycelium Differentiation and Antibiotic Production in Submerged Cultures of *Streptomyces coelicolor*. 74(12):3877-86
7. vanDiesel, D., Claessen, D., vanWezel G.P., (2014) Chapter One - Morphogenesis of *Streptomyces* in Submerged Cultures, *Advances in Appl. Microbiol.*, 89, 1-45
8. Zacchetti B., Smits P., Claessen D., (2018) Dynamics of Pellet Fragmentation and Aggregation in Liquid-Grown Cultures of *Streptomyces lividans*. *Front. Microbiol.* 9:943

3. Authors also correlate sampling points in relation to antibiotic oxytetracycline production. Which is clearly indicated in the title of the manuscript. However, it is not easy, based on the data presented in the text, what is the dynamics of oxytetracycline production in relation to the morphological development/pellet formation and glucose consumption. Therefore, if possible, I would suggest to present antibiotic titer on the Figure 1A, in addition to glucose concentration.

A: According to the reviewer's suggestion in addition to the glucose consumption we also present antibiotic titer in revised Figure 1A (please see below).

Fig. 1A

4. (R#2) As mentioned earlier, authors have completed very sound analysis of phosphoproteome accompanied with thorough analysis of the generated data. However, total proteome analysis did not really detect many proteins involved in biosynthesis of secondary metabolites. Is this expected? Could authors discuss this?

A: With thank the reviewer for this comment and in order to clarify it all proteins assigned by eggNOG mapper to functional category “Secondary metabolites biosynthesis, transport and catabolism” were highlighted in our proteome table for readers interested in this functional category (please see Data Set1, Sheet 1, proteins marked in grey). Note that ~ 300 proteins in the whole proteome of *S. rimosus* were assigned to this functional category and out of them we have detected 140 proteins. Nevertheless, we agree with the reviewer, only ~50% of proteins involved in biosynthesis of secondary metabolite (OTC) were detected during this study. However, despite the high concentration of glucose, the bacteria exhibited typical multiphase growth as well as the appearance of OTC at expected point of growth, during the growth arrest phase (for details please see explanation above – for Q2). OTC biosynthesis was at low levels and only 13/25 proteins involved in the biosynthetic pathway were detected in the proteome but not at statistically significant levels. Most of them (8/25) were detected during the SII and SIII phases when there is an increase in OTC in the medium. We are sorry if we did not clarify this but this information is now highlighted in the manuscript (see Table below; Note that these proteins are now marked in Supplementary, Data Set 1, Sheet3 as well). As stated above, for comparison, the high (50 g/L) concentration of glucose in this medium was standardized for low- and high-producers. It is probable that the higher concentration of glucose here repressed the expression of OTC production genes, which is likely why the levels of these gene products were so low in this proteomic analysis. Correlating with this, in the higher production strain (23383) when more glucose was utilized, 22/25 of the OTC pathway proteins were detected (ongoing study) – to be discussed

in a subsequent manuscript but below the list of these proteins is included in this response for information to this Referee.

Proteins identified in the qualitative analysis (OtcR, OtrB, OxyA, OxyB, OxyC, OxyD, OxyE, OxyF, OxyG/OxyH, OxyI, OxyJ, OxyK, OxyL, OxyM, OxyN, OxyP, OxyQ, OxyR, OxyS, OxyT, OtrA and OtcG) in the high OTC producer.

Proteins from OTC cluster in G7 (13/24):

Locus tag	Protein name	Significantly regulated in (with a 1.5 cut off on a log ₂ scale)
SRIM_038070	Methyltransferase, oxyT	rep 1 SIII, rep2 SII and SIII
SRIM_038075	hypothetical protein, oxyS	rep 1 SIII, rep2 SII and SIII
SRIM_038080	PPOX class F420-dependent oxidoreductase, oxyR	rep 1 SIII, rep2 SIII
SRIM_038100	cyclase family protein, oxyN	ratio not measured
SRIM_038105	SDR family oxidoreductase, oxyM	rep2 SIII
SRIM_038115	tetracycline nonaketamide D-ring cyclase/dehydratase OxyK	rep 1 SIII, rep2 SIII
SRIM_038120	SDR family NAD(P)-dependent oxidoreductase, oxyJ	ratio not measured
SRIM_038125	nuclear transport factor 2 family protein, oxyI	rep 1 SIII, rep2 SIII
SRIM_038130	AMP-binding protein, oxyG/oxyH	rep 1 SIII, rep2 SII and SIII
SRIM_038135	Methyltransferase, oxyF	rep 1 SIII, rep2 SII and SIII
SRIM_038150	acyl carrier protein, oxyC	ratio not measured
SRIM_038160	tetracycline polyketide synthase beta-ketoacyl synthase subunit OxyA	ratio not measured
SRIM_038170	oxytetracycline resistance efflux MFS transporter OtrB	not significantly increased

R#2 Although biosynthesis of oxytetracycline was detected during the fermentation, not even enzymes involved in biosynthesis of oxytetracycline were identified. On the other hand, a number of regulatory proteins, likely produced at very low concentrations were successfully detected. Therefore, considering that so few proteins catalyzing secondary metabolite biosynthesis were detected, it would be perhaps useful to discuss in the manuscript:

a) Is perhaps cultivation in the liquid culture, in a glucose rich medium reason for poor secondary metabolite production? How could this impact results on analysis of phospho(proteome)?

A: We hope that we have already answered the question about glucose and the media we are using. We also thank the reviewer for questions regarding proteins from the OTC cluster. It is obvious that we did not highlight them enough so they are also listed in the table we have shown for the answer to question no. 4. With respect to that we made some changes in the manuscript (please see lines 557-565) and we put information in which supplementary tables are the data of MS analyses related to proteins from OTC cluster. We have also highlighted them further in the supplementary tables themselves (please see lines 559 and 565).

The reviewer noted well that we discovered a lot of regulatory proteins (note that is was one of our major goals!). We assume that this can be attributed to the growth conditions and selected specific stages in which we collected the samples. As described above, the bacterium undergoes dynamic cellular processes during selected phases, in which cell lysis, pellet disintegration, and antibiotic synthesis occur continuously. In addition, some regulatory proteins control multiple cellular processes. For example, the predicted kinase, SRIM_017990 was found to be phosphorylated at all stages of growth during which time the

bacteria continuously synthesized OTC. However, this kinase is closely related to AfsK which is known to regulate polar growth, hyphae branching, and chromosome replication initiation in addition to antibiotic control (please lines 731-732).

b) Or is perhaps proteome analysis of enzymes involved in secondary metabolite production somehow limited?

A: Each bacterial sample can produce some products that limit the analysis. The authors applied modern technologies for MS analysis of bacterial (phospho) proteomes, but during this study *S. rimosus* samples proved to be very demanding due to the interference of other secondary metabolites (e.g. pigments, extracellular matrix ...). The approach applied to solve this problem is stated below (Q5). Note that during the final growth phase when the bacterium also produced the highest amount of antibiotic and when the OTC cluster genes should be most active, the bacterium also intensively produced an extracellular matrix that possibly interfered with protein extraction and detection levels.

It is certain that the technology and that the level of detection will improve over time, but we hope that our results (230 phosphoproteins of which 80% have not yet been reported in streptomycetes) will hopefully contributed to an increase in the bacterial database of phosphoproteomes.

R#2 Considering the title of the manuscript, it would be useful to add additional discussion related to this question and in correlation to points 2, 3 and 4.

A: As we have stated in the answers to the previous questions, in some parts of the manuscript we have added clarifications, additional references and additional information about the proteins involved in OTC biosynthesis (please see above).

5. Considering that the samples were selected at three time points, would it be possible to present more information on phosphorylation status of phosphorylated proteins at the different time points? It would be useful, if possible, to correlate this phosphorylation status by the onset of antibiotic oxytetracycline production. These kind of data would likely be very useful starting point for future studies.

A: We agree with the reviewer that measuring more points would most likely increase the number of detected (phospho) proteins. However, due to interference of secondary metabolites (pigment, cell-matrix proteins, extracellular polymers) MS measurements of *S. rimosus* samples turned out to be very demanding. As stated to Reviewer #1, a direct quantitative comparison of different growth phases was made using stable isotope (dimethyl) labeling. This allowed us to accurately quantify and at the same time minimize the number of missing values in the data set. In order to gain the best possible insight into the dynamics of (phospho)proteome, we selected three distinct growth phases in which significant regulatory changes occur and bacteria exhibit the most pronounced morphological changes (antibiotic

activation starts in (SI; for detail description of SI phase please above), bacteria enter into the stationary phase (SII) and late stationary phase (SIII)).

Note that Manteca and co-workers also performed MS analysis in three distinct phases of growth with the model organism (*S. coelicolor* – (phospho)proteomes) (1, 2).

In addition, it should be considered that the culture of *S. rimosus* is extremely heterogeneous. Especially during the SI/SII stages of growth, we have a population of older pellets that disintegrate and at the same time the young hyphae that still form new pellets. Consequently, a large number of proteins were found in both stages of growth. We believe the most interesting are proteins that were uniquely found in any of the selected stages and we focused on describing them.

Reference:

1. Manteca, A., Ye, J., Sánchez, J., Jensen, O.N., (2011) Phosphoproteome analysis of Streptomyces development reveals extensive protein phosphorylation accompanying bacterial differentiation. J Proteome Res., 10(12):5481-92
2. Rioeras, B., Shliaha, P.V., Gorshkov, V., Yagüe, P., López-García, M.T., Gonzalez-Quíñonez, N., Kovalchuk, S., Rogowska-Wrzesinska, A., Jensen, O.N., Manteca, A., (2018) Quantitative Proteome and Phosphoproteome Analyses of Streptomyces coelicolor Reveal Proteins and Phosphoproteins Modulating Differentiation and Secondary Metabolism. Mol Cell Proteomics, 17(8):1591-1611

Minor comments:

R#2 Page 3, line 61: oxytracycline

A: We thank the reviewer for noticing this typo, it has been changed to „oxytetracycline“ (line 52).

R#2 Figure S2 (to explain better the figure- outer/inner cycle)

R#2: Figure S3 (to explain better the figure- outer/inner cycle)

A: We thank the reviewers for noticing the lack of images description. Note that these figures are now **Fig. S1A and B** and corresponding text is added to Fig. legends:

Figure S1. The distribution of identified (phospho)proteins into major functional categories in *S. rimosus* and *S. coelicolor*. The outer circles display the main functional categories obtained by eggNOG-mapper for *S. rimosus* (**A**) and *S. coelicolor* (**B**) proteomes while inner circle those identified for phosphoproteomes. Only (phospho)proteins with one assigned functional category are shown (i.e. approx. 93% of the proteome and 90% of the phosphoproteome for *S. rimosus* and approx. 95% of (phospho)proteome for *S. coelicolor*).

A request to change:

- **Figure S5 / now Figure S3:** In addition to answering reviewers' questions and the changes we have made to the manuscript, the authors would like to change the colours in the Figure S5 (please see below) to facilitate the comparison of distributions of functional categories and their subcategories in different clusters.

Namely, in the revised figure different shades of one color denote subcategories that belong to one specific category. In this way, we have simplified the graphical presentation of the results and we hope that the reviewers and editor will agree with this change.

Figure S3. Distribution of functional subcategories in each cluster. Each bar represents the distribution of functional subcategories in each cluster of proteins with similar temporal abundances.

- In addition, since the number of figures in supplements is limited to 10, the authors request permission to remove **Fig. S1** (the Western blot) because it is a control experiment to which reviewers had no objection, as well as **Fig. S6** which is an interesting observation but not crucial for results in this manuscript.
- Please note that, we added some additional results to Fig. 1A. Consequently, we had to expand some specific parts in the text with a corresponding description of the results and add appropriate discussion. For this reason we have shortened other parts of the manuscript to the maximum, as can be seen in the marked version. All changes requested by reviewers are marked in red while other changes in the text are marked by Track Changes.

July 13, 2022

Prof. Dušica Vujaklija
Ruđer Bošković Institute
Molecular Biology
Bijenicka 54
Zagreb 10000
Croatia

Re: mSystems00199-22R2 (Phosphoproteome dynamics of *Streptomyces rimosus* during submerged growth and antibiotic production)

Dear Prof. Dušica Vujaklija:

Your manuscript has been accepted, and I am forwarding it to the ASM Journals Department for publication. For your reference, ASM Journals' address is given below. Before it can be scheduled for publication, your manuscript will be checked by the mSystems production staff to make sure that all elements meet the technical requirements for publication. They will contact you if anything needs to be revised before copyediting and production can begin. Otherwise, you will be notified when your proofs are ready to be viewed.

Publication Fees:

If you would like to submit a potential Featured Image, please email a file and a short legend to msystems@asmusa.org. Please note that we can only consider images that (i) the authors created or own and (ii) have not been previously published. By submitting, you agree that the image can be used under the same terms as the published article. File requirements: square dimensions (4" x 4"), 300 dpi resolution, RGB colorspace, TIF file format.

We recognize that the video files can become quite large, and so to avoid quality loss ASM suggests sending the video file via <https://www.wetransfer.com/>. When you have a final version of the video and the still ready to share, please send it to mSystems staff at msystems@asmusa.org.

Sincerely,

Joshua Elias
Editor, mSystems

Journals Department
Fig. S1: Accept

Table S3: Accept

Table S2: Accept

Fig. S2: Accept

Fig. S3: Accept

Table S4: Accept

Table S5: Accept

Data Set S1: Accept

Table S1: Accept